# TotalRecall: A Bidirectional Candidates Generation Framework for Large Scale Recommender & Advertising Systems

## Abstract

Recommender (RS) and Advertising/Marketing Systems (AS) play the key roles in E-commerce companies like Amazon and Alibaba. RS needs to generate thousands of item candidates for each user ($u2i$), while AS needs to identify thousands or even millions of high-potential users for given items so that the merchant can advertise these items efficiently with limited budget ($i2u$). This paper proposes an elegant bidirectional candidates generation framework that can serve both purposes all together. Specifically, our framework finds its practical applications on generating millions of users for a large number of items, and the marketing scenarios of small and medium enterprises (SMEs) with cloud-service. Besides, our framework is also superior in these aspects: $i$). We theoretically show that our modeling objective is equivalent to the modeling objectives of the SOTA algorithms if measured by the metrics of $u2i$ and $i2u$. $ii$). Our framework can easily incorporate many DNN-architectures, and achieve SOTA results with the same architecture. $iii$). We empirically show that our framework can diversify the generated candidates, and ensure fast convergence.

## 1 Introduction

In the Internet era, E-commerce companies usually act as a platform to connect both users and merchants. They can collect huge amount of data of users' behavior on merchants' items, and utilize the data to serve them better, for example, to build better search/recommender/advertising systems.

When users browse the E-commerce websites/apps, they usually want to buy some desired items, or find something interesting, and that's what the search engine and recommender system (**RS**) are built for. Besides, the merchant usually wants to mine the potential users of its items, and then starts a campaign to prompt those users to buy the items. The advertising system has the functionality to help the merchant, and we may call it the potential users mining system (**PUMS**).

In an industrial large-scale RS, there are usually two stages, the candidates generation (**CG**) stage and the ranking stage (Covington et al., 2016). CG plays a very import role as it sets the upper bound of the recommender. A single CG method usually lacks diversity, so the industrial CG system usually consists of different subsystems to generate candidate items, for example, $i$). hot items, $ii$). items similar to users (Covington et al., 2016; Li et al., 2019), $iii$). items similar to clicked items (Linden et al., 2003), $iv$). items clicked by similar users (Wang et al., 2006).

The PUMS has been studied and applied in marketing community for many years (Bult & Wansbeek, 1995; Kim et al., 2005), and recently it is also introduced to social network to attract potential customers of companies (Pennacchiotti & Popescu, 2011; Pang et al., 2013). Similarly in E-commerce companies, it can be used to discover potential customers for merchants, brands and etc. Although it is widely used, people usually treat it as a simple binary classification problem, i.e., to predict whether a given user is the potential user or not. Models like LR, SVM, XGBoost and etc. (Hosmer Jr et al., 2013; Chang & Lin, 2011; Chen & Guestrin, 2016) are usually employed.

In this paper, we unify the two systems in our TotalRecall (**TR**) framework, and apply it in both large-scale PUMS and marketing scenarios of SMEs with the cloud-service. Our main contributions are listed below:

- **Bidirectional Candidates Generation**: TR can be trained once and then used to generate candidate items ($u2i$) and users ($i2u$) all together without any additional operations. Theoretically we show that it is due to the modeling of joint probability of $u$ and $i$.

- **Achieving SOTA results in both RS and PUMS**: We compare to two different algorithms, i.e., Matrix Factorization (**MF**) with Bernoulli distribution and sequential modeling and MF with Multinomial distribution, and achieve comparable or better results than previous SOTA.

- **Equivalence of the algorithms based on Bernoulli and Multinomial distributions**: To the best of our knowledge, we are the first to show that modeling the user-item interaction with Bernoulli and Multinomial distributions are equivalent.

- **Fast Convergence, and Diversified Candidates**: TR converges much faster compared to MF models, up to $16\times$ times. Our experiments show that TR can increase the diversity of the generated candidates.

## 2 RELATED WORK

### 2.1 RECOMMENDER SYSTEM (RS)

RS has been studied for many years, and in the early years, Collaborative Filtering (**CF**) and its variants (Su & Khoshgoftaar, 2009) are widely adopted. Later its descendant MF (Funk) is proposed to solve the problem more elegantly with higher accuracy, and Probabilistic Matrix Factorization (**PMF**) (Mnih & Salakhutdinov, 2008) builds a solid theoretic foundation for MF based on probability theory.

Although MF performs very well in many datasets (Rendle et al., 2020), it is usually not adopted in industry. The modern industrial RS can involve up to billions of users and hundreds of millions of items, and consist of multiple stages, e.g., CG and ranking stages (Covington et al., 2016).

Various deep neural networks are adopted to tackle the complicated system, for example, in the CG stage, the problem is formulated as the multi-class classification, and users' and items' representations are detached with the two-tower architecture to balance the efficiency and accuracy in online serving (Covington et al., 2016; Li et al., 2019; Cen et al., 2020).

In the ranking stage, the problem is formulated as the binary classification, and various models has been developed: wide & deep model (Cheng et al., 2016) captures explicit and implicit feature crossing, Deep Interest Network (Zhou et al., 2018) and Deep User Perception Network (Ni et al., 2018) utilize attention mechanism (Bahdanau et al. (2014)) to capture users' short-term interests.

### 2.2 POTENTIAL USERS MINING SYSTEM (PUMS)

PUMS mines the potential users for given items. The 'item' could be anything that users can interact with, for example, an insurance product (Kim & Street, 2004), a company/business (Pennacchiotti & Popescu, 2011; Pang et al., 2013; Lo et al., 2016), a specific information (e.g., tweet) in social medias (Tang et al., 2015; Gui et al., 2019) and even another user (Guy, 2018). These problems are usually treated as binary classification and solved with tools like SVM, LR and etc. for each individual item independently (Hosmer Jr et al., 2013; Chang & Lin, 2011; Chen & Guestrin, 2016).

In the E-commerce company like Amazon, the 'item' could be a product, a brand, a product category, a merchant and etc, and the number of the 'items' range from thousands to hundreds of millions. It will be impractical to model each item individually, so we usually model the items all together using binary classification similar to Rendle et al. (2020), which will be stated in detail later in Sec. 5.

## 2.3 SAMPLED-SOFTMAX AND INFONCE LOSS

In the CG stage of RS, sampled-softmax (**SSM**) loss (Jean et al., 2015) is widely used with different background distributions, such as log-uniform distribution (Powers, 1998), empirical marginal distribution (Yi et al., 2019) and mixed distribution (Yang et al., 2020).

In contrastive learning, InfoNCE loss is proposed in Oord et al. (2018) to learn representations through self-supervision. Later it is widely adopted in CV (Chen et al. (2020)) and NLP (Gao et al. (2021)) and performs very well.

InfoNCE and SSM losses are both trying to solve multi-class classification of very large vocabulary. They both use negative sampling and can be written in the same analytical form:

$$l = \frac{1}{|\mathcal{D}|} \sum_{(x,y) \in \mathcal{D}} - \log \frac{\exp(f_\vartheta(x, y) - \log p_n(y|x))}{\exp(f_\vartheta(x, y) - \log p_n(y|x)) + \sum_{y' \in \mathbb{Y}_x} \exp(f_\vartheta(x, y') - \log p_n(y'|x))},$$

(1)

where $(x, y) \in \mathcal{D}$ is the training data, and usually $x$ is the input features and $y$ is the class, $f_\vartheta(x, y)$ is the output of the neural network parametrized by $\vartheta$, and $\mathbb{Y}_x$ contains hundreds or thousands of negative samples from background distribution $p_n(y|x)$. The main differences are that the $p_n(y|x)$ in SSM can be arbitrarily chosen, while in InfoNCE it is the empirical marginal distribution $\hat{p}_{\text{data}}(y)$.

Recently, InfoNCE loss is exploited in RS to optimize $\frac{p(y|x)}{p(y)}$ instead of $p(y|x)$ for the debiasing purpose, and it can be easily implemented with in-batch negative sampling and features of $y$ can be fully utilized (Zhou et al., 2021).

# 3 METHOD

## 3.1 PROBLEM DEFINITION

In E-commerce companies, when a user $u$ clicked an item $i$ at time $t$, a record $(u, i, t)$ was logged as the implicit feedback.

Given the raw logs $\{(u, i, t)\}$, there are usually two ways to process the data: one is to directly create a user-item interaction matrix $\boldsymbol{S}_{ui}$ with the value $s_{u,i} = 1$ if the $(u, i)$ pair appears in the log (Su & Khoshgoftaar (2009)); the other is to formulate the problem as next-item prediction, i.e., from the raw log we create a dataset $\mathcal{D} = \{(x_{u,t}, y_{u,t}) : t \in \{1, 2, ..., T_u\} | u \in \{1, 2, ..., N\}\}$, where $x_{u,t} = \{y_{u,1:(t-1)}\}$ represents $u$'s clicks prior to $t$-th click $y_{u,t}$ and $T_u$ is the number of clicks of $u$ (Covington et al. (2016)). We define the click sequence $x_{u,t}$ as the pseudo-user, so we can create a pseudo-user-item interaction matrix from $\mathcal{D}$, and all possible click sequences form the pseudo-user set.

Therefore, they both deal with the 2D interaction matrix $\boldsymbol{S}_{ui}$, irrespective of whether the $u$ is a real user or pseudo-user or not. In the remaining of the paper, we will not distinguish between user and pseudo-user, and only focus on solving the $\boldsymbol{S}_{ui}$.

$\boldsymbol{S}_{ui}$ consists of all users as the row and all items as the column, and its entries are either 1 or unknown. Let's assume all users form the set $\mathbb{U} = \{u_1, u_2, ..., u_M\}$, and all items form the set $\mathbb{I} = \{i_1, i_2, ..., i_K\}$. Given a user $u \in \mathbb{U}$, we will generate candidate items from the item pool $\mathbb{I}$ (RS, $u2i$), while given an item $i \in \mathbb{I}$, we shall find out the potential users out of all users $\mathbb{U}$ (PUMS, $i2u$).

The candidates generation of RS and PUMS are of the opposite direction, and traditionally they are solved separately. But essentially they are both trying to estimate some kind of probabilities of the unknown entries in $\boldsymbol{S}_{ui}$, so we propose to model them jointly.

## 3.2 RECOMMENDER SYSTEM

In RS, there are usually two types of modeling objectives when dealing with $\boldsymbol{S}_{ui}$, and we briefly introduce them in the following sections (See App. A.1 for more details).

### 3.2.1 GAUSSIAN AND BERNOULLI DISTRIBUTION

The $u$-$i$ interaction score $s_{u,i}$ of $\boldsymbol{S}_{ui}$ is treated as the random variable, and assumed to be i.i.d. following a certain distribution, for example, Gaussian (Mnih & Salakhutdinov (2008)) or Bernoulli (Johnson (2014); He et al. (2017); Rendle et al. (2020)) distributions.

### 3.2.2 MULTINOMIAL DISTRIBUTION

Here the modeling objective is the random variable $\boldsymbol{s}_u$, defined as the interacting item of $u$. For example, given the total number of interactions $N_u = \sum_{i \in \mathbb{I}} s_{u,i}$ of $u$, $\boldsymbol{s}_u$ is assumed to be sampled from a multinomial distribution $\mathrm{Mult}(N_u, \boldsymbol{p}_u)$, where $\boldsymbol{p}_u$ is a $K$-dimensional vector whose elements are the probability and sum to 1 (Liang et al., 2018). Although not explicitly stated in many papers (Covington et al., 2016; Yi et al., 2019; Li et al., 2019; Cen et al., 2020), the multinomial distribution is actually assumed.

The modeling objectives in Sec. 3.2.1 and 3.2.2 are quite different, as the former is $s_{u,i}$ and the latter is $\boldsymbol{s}_u$. In MF, we usually model $s_{u,i}$, while in the sequential modeling of the CG stage in industry, we model $\boldsymbol{s}_u$. Later in Sec. 3.5, we will show that modeling $\boldsymbol{s}_u$ and $s_{u,i}$ can be theoretically equivalent, and it is verified by the experiments in Sec. 4.1 and 5. However, modeling $\boldsymbol{s}_u$ can boost the converging speed a lot.

### 3.3 POTENTIAL USERS MINING SYSTEM

The PUMS seems to be the twin system of RS, and similarly we could also define two modeling objectives $s_{u,i}$ and $\boldsymbol{s}_i$, where $\boldsymbol{s}_i$ denotes the user who interacts with the item $i$. $s_{u,i}$ and $\boldsymbol{s}_i$ are assumed to follow Bernoulli and Multinomial distributions respectively. In Sec. 5 we will compare the two methods.

### 3.4 TOTALRECALL: UNIFICATION OF RS AND PUMS

Motivated by Sec. 3.2 and 3.3, we propose to model the two systems jointly. Specifically, the optimization objectives $\boldsymbol{s}_u$ and $\boldsymbol{s}_i$ in Sec. 3.2.2 and 3.3 are formulated in one framework and optimized together. Later we show that by properly choosing the model architecture and loss function, we are treating $u$ and $i$ equally and modeling the joint probability.

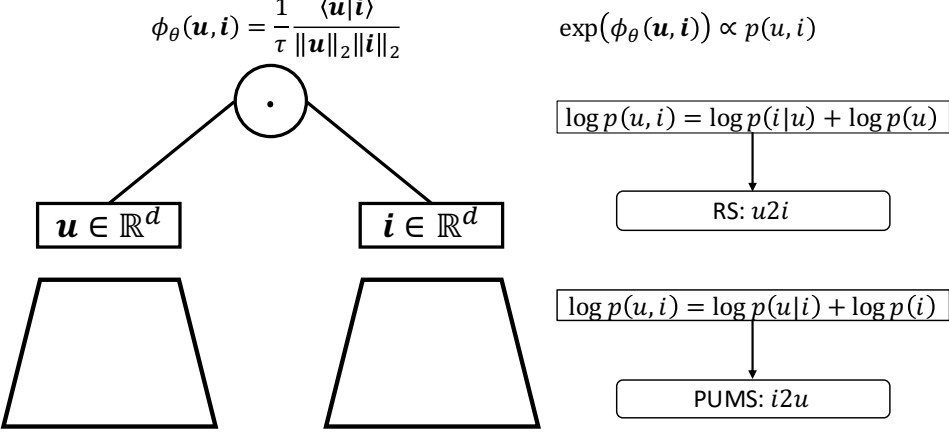

Figure 1: Model architecture of TotalRecall. Our framework will make to dot product $\phi_\theta(u, i)$ converge to the logarithm of the joint probability $p(u, i)$, and thus can be directly applied in RS to emulate the conditional probability $p(i|u)$, and also in PUMS to replace $p(u|i)$.

### 3.4.1 MODEL ARCHITECTURE

We choose the classical two-tower architecture (Huang et al., 2013; Covington et al., 2016; Rendle et al., 2020) for two reasons: one is that users and items can be treated equivalently, and the other is

that no feature crossing occurs before the final logits (See Fig. 1). So users' and items' embeddings can be inferred separately, and then approximate nearest neighbor (ANN) (Liu et al., 2004) search algorithm can be applied in online serving.

The output of two towers are $d$-dimensional vectors $\boldsymbol{u} = f_\theta(u) \in \mathbb{R}^d$ and $\boldsymbol{i} = g_\theta(i) \in \mathbb{R}^d$, where $\theta$ is the model parameter. The dot product $\langle \boldsymbol{u} | \boldsymbol{i} \rangle$, or the function of it is used as the sufficient statistics of the probability distributions defined in Sec. 3.2 and 3.3 (Check App. A.1 for details). We find that l2-normalizing $\boldsymbol{u}$ and $\boldsymbol{i}$ and then rescaling the dot product by the temperature $\tau$ will lead to better and robust results:

$$\phi_\theta(\boldsymbol{u}, \boldsymbol{i}) = \frac{1}{\tau} \frac{\langle \boldsymbol{u} | \boldsymbol{i} \rangle}{||\boldsymbol{u}||_2 ||\boldsymbol{i}||_2}. \tag{2}$$

### 3.4.2 LOSS FUNCTION

We propose the Bi-InfoNCE loss:

$$l = \frac{1}{|\boldsymbol{S}_{u,i}|} \sum_{u \in \mathbb{U}, i \in \mathbb{I}, s_{u,i}=1} - \log \frac{\exp(\phi_\theta(u,i) - \log q(i))}{\exp(\phi_\theta(u,i) - \log q(i)) + \sum_{i' \in \mathbb{I}_u} \exp(\phi_\theta(u,i') - \log q(i'))}$$
$$- \log \frac{\exp(\phi_\theta(u,i) - \log \hat{q}(u))}{\exp(\phi_\theta(u,i) - \log \hat{q}(u)) + \sum_{u' \in \mathbb{U}_i} \exp(\phi_\theta(u',i) - \log \hat{q}(u'))}, \tag{3}$$

where $\mathbb{I}_u \subset \mathbb{I}$ and $\mathbb{U}_i \subset \mathbb{U}$ contain hundreds or thousands of in-batch negative samples, and $q(i)$ and $\hat{q}(u)$ are empirical marginal distribution calculated using the training data $\boldsymbol{S}_{ui}$. It can be shown that optimizing this loss will make $\phi_\theta(u,i)$ converge to $\log \hat{p}_{\text{data}}(u,i)$ (See App. A.3 for details.).

If we only optimize the first part of Eq. 3, $\phi_\theta(u,i)$ will converge to $\log \hat{p}_{\text{data}}(i|u)$, and optimizing only the second part will let $\phi_\theta(u,i)$ converge to $\log \hat{p}_{\text{data}}(u|i)$. We refer to the two cases as Uni-InfoNCE losses.

### 3.5 EQUIVALENCE BETWEEN MODELING WITH BERNOULLI AND MULTINOMIAL DISTRIBUTIONS

As discussed in the previous sections, we can use both Bernoulli and Multinomial distributions to model the same user-item interaction matrix $\boldsymbol{S}_{ui}$, and usually their results are comparable in both RS and PUMS (See Sec. 4.1 and 5).

The apparent connection between the two algorithms motivates us to explore their relations, and we prove that they are theoretically equivalent on modeling the user-item interaction matrix. Specifically, different negative sampling methods of Bernoulli distribution correspond to different objectives modeled with Multinomial distributions.

For example, in MF with Bernoulli distribution ($\text{MF}_b$), we could have 5 different negative sampling methods, i.e., for a given positive $(u,i)$ pair, we sample the negative pairs in these ways: 1). '$\text{MF}_b$ [0:8]'[1], we randomly sample 8 $i$s from the item set $\mathbb{I}$, and construct 8 negative pairs with the given $u$; 2). '$\text{MF}_b$ [8:0]', we sample 8 $u$s from the user set $\mathbb{U}$, and construct 8 negative pairs with the given $i$; 3). '$\text{MF}_b$ [4:4]', we sample 4 $u$s and 4 $i$s, and construct 8 negative pairs with the given $i$ and $u$ respectively; 4). similar to 3), but the negative samples are doubled; 5). '$\text{MF}_b$ [8]', we sample 8 $(u,i)$ pairs uniformly from the $M \times K$ pairs with $i \in \mathbb{I}$ and $u \in \mathbb{U}$ as the negative pairs.

We prove that 1) is equivalent to modeling $p(i|u)$, 2) to $p(u|i)$, 3) and 4) to $\frac{p(u,i)}{p(u)p(i)}$, and 5) to $p(u,i)$. The details can be found in App. A.4.

### 3.6 MOTIVATION AND PRACTICAL APPLICATIONS OF TR

TR is developed to solve two practical problems: one is the large-scale potential users mining, and the other is the marketing in SMEs.

---

[1]This sampling method is actually used in $\text{MF}_b$ in RS by Rendle et al. (2020)

Table 1: Baselines from Dacrema et al. (2021) and Rendle et al. (2020) and our results. The best results are highlighted in bold, the second best result is underlined.

| Method | Movielens | | Pinterest | | Result from |
|---|---|---|---|---|---|
| | HR@10 | NDCG@10 | HR@10 | NDCG@10 | |
| Popularity | 0.4533 | 0.2542 | 0.2740 | 0.1409 | Dacrema et al. (2021) |
| SLIM | 0.7162 | 0.4468 | 0.8679 | 0.5601 | Dacrema et al. (2021) |
| iALS | 0.7109 | 0.4382 | 0.8761 | 0.5590 | Dacrema et al. (2021) |
| MLP+GMF | 0.7093 | 0.4349 | 0.8777 | 0.5576 | Dacrema et al. (2021) |
| $MF_b$ by Rendle | **0.7294** | 0.4523 | 0.8895 | 0.5794 | Rendle et al. (2020) |
| $MF_b$ (ours) | 0.726 | 0.4505 | 0.8925 | 0.5838 | - |
| TR (ours) | 0.7281 | **0.4541** | **0.8936** | **0.5866** | - |

For example, we have 1 billion users and 10,000 items, and we need to mine the top 1 million users for each of the items. This problem can be easily solved with TR: we infer the users' and items' embeddings with TR, and then apply the ANN search to find the top 10 items for each user. Now we have 10 billion $(u, i)$ pairs and their similarity scores, and we just pick the top 1 million for each item based on the same scores. This method also indirectly ensures that each person will receive at most 10 advertisement messages.

This can only be achieved if we model the joint probability $p(u, i)$. We also list some alternative methods to solve the problem in App. A.5, but they are either not good in terms of accuracy or not practical in terms of computation and storage costs, efficiency, risk and maintainability. TR has been deployed in our production environment to serve this purpose.

In the marketing scenario of SMEs, two strategies are widely adopted: $i$). SMEs pick some specific groups of users, and then choose the top products for each user and send the personalized campaign messages. This can be regarded as the RS. $ii$). SMEs have some new or special products and want to find the target users. This is the PUMS.

The two marketing requirements can be completed either with two separate models or with the unified model TR. Due to the limited computation and storage budgets, and the shortage of talents, TR is the best choice for them. Actually, TR is already implemented in one of our cloud-service products for SMEs.

## 4 EXPERIMENTS: RS

As stated in Sec. 3.2, two types of modeling objective are well studied in literature, and usually they use different datasets and baselines. So we compare our results with them separately using different datasets. The details of the datasets and processing methods can be found in App. A.6.

### 4.1 MF WITH BERNOULLI DISTRIBUTION

We compare TR to $MF_b$ in Rendle et al. (2020), as described in Sec. 3.2.1. Our implementation of $MF_b$ is a bit different from Rendle et al. (2020): we implement it using tensorflow (Abadi et al., 2016) with Adam optimizer (Kingma & Ba, 2014).

We tried L2 regularization but failed to obtain the results reported by Rendle et al. (2020). We also tried dropout (Srivastava et al., 2014), but it was unstable and need to be re-tuned for different dimensions $d$. Finally we find that the $\phi_\theta(\boldsymbol{u}, \boldsymbol{i})$ defined in Eq. 2 can actually act as a kind of regularization, so we use it rather than the one defined by Eq. (2) in Rendle et al. (2020).

We use Movielens-1m and Pinterest datasets, and the evaluation metrics are HiteRate@10 and NDCG@10. $MF_b$ and TR have the same two-tower architecture and same number of parameters, and the key differences are the training data and the optimizing objectives: $MF_b$ uses explicit negative sampling, i.e., $MF_b$ [0:8], and optimize the binary cross-entropy loss, and TR uses implicit in-batch negative sampling and Bi-InfoNCE loss.

Table 2: For Amazon books data, we use $\tau = 0.1$, batch-size $= 128$. For Taobao data, we use $\tau = 0.067$, batch-size $= 2048$. ComiRec-SA and ComiRec-DR are two multi-interest models proposed in Cen et al. (2020). The best results are highlighted in bold, the second best result is underlined.

| | Amazon Books Metrics@50 | | Taobao Metrics@50 | |
|---|---|---|---|---|
| | Recall | Hit Rate | Recall | Hit Rate |
| Popular | 2.400 | 5.226 | 0.735 | 9.309 |
| YouTube [SSM w/o l2-norm] | 7.312 | 15.894 | 6.172 | 39.108 |
| ComiRec-SA | 8.467 | 17.202 | 9.462 | 51.064 |
| ComiRec-DR | 8.106 | 17.583 | 9.818 | 52.418 |
| Youtube [SSM with l2-norm] | 9.37 (+28%) | 19.25 (+21%) | 6.9 (+12%) | 42.87 (+10%) |
| TR-Youtube [Uni-InfoNCE $u2i$] | 10.34 (+41%) | 21.15 (+33%) | **7.15 (+16%)** | **43.79 (+12%)** |
| TR-Youtube [Bi-InfoNCE] | 10.12 (+38%) | 20.554 (+29%) | 7.14 (+16%) | 43.57 (+11%) |
| TR-Youtube [next-7-prediction] | **11.571 (+58%)** | **23.221 (+46%)** | - | - |

The results are shown in Tab. 1. In general, $MF_b$ and TR are comparable, and this is consistent with our theory in Sec. 3.5, because they are both modeling $p(i|u)$ or $p(u,i)$ ultimately. Besides, the diversity of TR is also larger than $MF_b$ as shown in Fig. 3. (Check App. A.6.1 for more details.)

When measuring the training efficiency, TR converges to the optima up $16\times$ times faster than $MF_b$ (See Fig. 3 in Appendix). We attribute it to two reasons: $i$). For $MF_b$ [0:8] used here, 9 processed samples will consist of only 1 positive samples on average. While in Bi-InfoNCE (or Uni-InfoNCE / SSM), 9 processed samples are all positive samples. $ii$). For $MF_b$, each sample is either positive or negative, and the binary-cross-entropy loss allows it to compare with itself only. In TR, Bi-InfoNCE (or Uni-InfoNCE / SSM) loss will compare each sample with all the other samples in the batch.

Bearing in mind that $MF_b$ and Bi/Uni-InfoNCE can theoretically converge to the same optimum, the qualitative explanation can provide some intuitive understandings of the fast convergence. The fast converging speed makes it more practical to use TR in industrial applications with huge amount of data.

### 4.2 SEQUENTIAL MODELING WITH MULTINOMIAL DISTRIBUTION

We compare TR to the sequential modeling with Multinomial distribution, as described in Sec. 3.2.2. This modeling method is usually used in the CG stage of industrial large-scale RS, because it can deal with huge amount of data very efficiently, and yield good results.

For simplicity, we use the popular Youtube DNN (Covington et al., 2016) implemented by Cen et al. (2020) as the backbone architecture of TR. Other complicated architectures can be easily incorporated into TR framework.

We use the same datasets as Cen et al. (2020) for a fair comparison, i.e., Amazon books data [2] and Taobao click data in Tianchi competition [3]. The data is split into train, validation and test data. The training data is usually processed as next-item prediction as in Sec. 3.1, and we extend it into next-$n$-item prediction, i.e., instead of predicting only the next item, we predict all the next $n$ items. The results are shown in Tab. 2. More results are shown in Fig. 4 and 5 in Appendix.

Our experiments show that proper regularization using l2-normalization and the scaling parameter $\tau$ can greatly improve the results, as it confines $\phi_\theta(u,i)$ inside $[-1/\tau, 1/\tau]$. Better training data due to next-$n$-prediction can also improve the results, because it alters the distribution of the training data so that it gets close to the test data (See Fig. 6 in Appendix).

Besides, we have three observations. $i$). **Comparable results**: TR's result is comparable to the SOTA results from SSM and Uin-InfoNCE. This shows that the modeling objectives $p(u,i)$ and

---

[2] http://jmcauley.ucsd.edu/data/amazon/index.html
[3] https://tianchi.aliyun.com/dataset/dataDetail?dataId=649&userId=1

$p(i|u)$ perform similarly if measured by the metrics for $u2i$ candidate generation. $ii$). **Improved diversity**: We try SSM with different background distributions, and compare with Uni-InfoNCE and Bi-InfoNCE, and find that Bi-InfoNCE gives better overall diversity as measured by the related metrics (See Fig. 4 in Appendix). We don't have a rigorous explanation and would like to leave it for the future study. $iii$). **Faster convergence**: Compared to SSM or Uni-InfoNCE without l2-normalization, adding l2-normalization or using Bi-InfoNCE will make the models learn faster (See Fig. 4 in Appendix). This is because l2-normalization will restrict the vector space that $u$ and $i$ could explore, and Bi-InfoNCE utilizes the data more efficiently compared to Uni-InfoNCE, i.e., the utilization rate is doubled.

## 5 EXPERIMENTS: PUMS

In E-commerce, there are usually two practical scenarios of $i2u$ candidates generation: one is to mine the potential users for a single item, and the other is to find users for a group of similar items. Because the marketing staff or some merchants regularly pick a group of similar items, and set the specific campaigns to advertise them.

Previously, the problem is usually formulated as the binary classification for each individual case, but it is not generally applicable because the number of items or groups could be quite large, and it is impractical to build so many models.

One solution is to merge all the binary classification models into one, just like $\text{MF}_b$ in Sec. 3.2.1 and 3.3. The other is our TR framework.

### 5.1 DATASET AND EVALUATION METRICS

We use Movielens-1m because it is also used in $\text{MF}_b$ of RS, and it can be used to mine the potential users of both the single item and the group of similar items.

Movielens-1m records 6040 users and 3706 movies. The data is split into training, validation and test data, and we use the validation data to select the proper model configurations. The final results are reported on the test data.

We use precision@10 and recall@10 to evaluate the accuracy of different algorithms, i.e., we mine the top 10 users for a single item or a group of items, and then evaluate the precision and recall. We count the distinct users of the overall results to measure the diversity. (See App. A.7 for details about the dataset and metrics.)

### 5.2 POTENTIAL USERS OF A SINGLE ITEM

We first select movies which have been watched by at least 50 users, and it results in 2514 movies. For each of these movies, we randomly pick 1% of its users to form the valid and test datasets, with half of the movies as the validation data and the other half as the test data (See App. A.7.). The remaining data is used to train the model, and the validation data is for selecting the best hyper-parameters, and the results are reported on the test data.

The results are shown in Tab. 3. 'TR [Bi-InfoNCE]' achieves the best results in terms of the accuracy and diversity. High diversity implies that the advertisements can reach more users, even the non-active users, thus alleviate the advertising fatigue.

Besides, TR uses the same training data and model configurations for both RS and PUMS, and achieves comparable in both cases. The learned $u/i$ representations can be applied to the two systems without any additional modifications. In contrast, $\text{MF}_b$'s modeling objective (Eq. 4) seems to treat $u$ and $i$ equally, but the outcomes depend highly on the negative sampling methods (See Tab. 3). This is consistent with our theoretical analysis in Sec. 3.5.

Last but not least, TR learns $16\times$ times faster than $\text{MF}_b$ (See Fig. 7 in Appendix). The reason is the same as in Sec. 4.1.

Our study of mining potential users of a group of items can be found in App. A.7.2.

Table 3: Overall performance of mining the potential users of a single item. The % is omitted for precision and recall. The hyper-parameters of $MF_b$ are batch-size $= 4096$ and $\tau = 0.125$, and for TR they are batch-size $= 512$ and $\tau = 0.125$. We use $d = 96$ for all models for a fair comparison. The diversity values of models with very low accuracy are omitted.

| model | precision@10 | recall@10 | distinct users |
|---|---|---|---|
| $MF_b$ [0:8] | 1.47 | 5.61 | - |
| $MF_b$ [8:0] | 4.44 | 12.82 | 2046 |
| $MF_b$ [4:4] | 3.12 | 9.28 | - |
| $MF_b$ [8:8] | 3.02 | 8.64 | - |
| TR [Uni-InfoNCE $u2i$] | 1.19 | 4.51 | - |
| TR [Uni-InfoNCE $i2u$] | 4.35 | 12.78 | 2064 |
| TR [Bi-InfoNCE] | **4.58** | **13.82** | **2202** |

## 6 CONCLUSION

In this paper, CF and the sequential modeling (i.e., next-item prediction) of RS, and PUMS are abstracted into a unified $u$-$i$ interaction matrix $\boldsymbol{S}_{ui}$, and thus are treated equivalently as deriving some probabilities for the unknown entries from the known entries of $\boldsymbol{S}_{ui}$. We also show their connections and differences based on probability theory.

We prove that modeling $p(s_{u,i} = 0/1|u,i)$ with different sampling methods can be equivalent to modeling $p(i|u)$, $p(u|i)$, $p(u,i)$ and $\frac{p(u,i)}{p(u)p(i)}$.

We propose the TR framework to solve $u2i$ and $i2u$ CG all together. In TR, $u$ and $i$ are treated equally, and the joint probability $p(u,i)$ is actually derived, instead of the conditional probability $p(i|u)$ or $p(u|i)$. Therefore, the learned representations of $u$ and $i$ in TR can be used in both $u2i$ and $i2u$ CG.

Our extensive experiments show that TR can achieve comparable results in each domain measured by the metrics for either $u2i$ or $i2u$ candidates generation. TR learns $16\times$ times faster than $MF_b$. Besides, TR generates diversified candidates.

In the future, we plan to study the vector space of TR in depth, and try to unify different objects and users into one space, for example, products, merchants, brands, categories of products. In such a space, everything is linked, and candidates generation can be applied on any pair of objects.

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

# A APPENDIX

## A.1 DISTRIBUTIONS

In Sec. 3.1 we define the interaction matrix $\boldsymbol{S}_{ui}$, and it has the known entries with value 1 and unknown entries. Here we define a corresponding matrix $\boldsymbol{P}$ of the same shape, and its entries are the probabilities estimated by a certain algorithm.

In Sec. 3.2.1, the corresponding probability matrix $\boldsymbol{P}_{\text{MF}}$ is defined as follows:

$$p_{u,i} := p(s_{u,i} = 1 | u, i), \tag{4}$$

where $\sum_{s_{u,i} \in \{0,1\}} p(s_{u,i}|u,i) = 1$.

Here we can assume $s_{u,i}$ follows Gaussian distribution: $s_{u,i} \sim \mathbf{N}(\phi(\boldsymbol{u}, \boldsymbol{i}), \hat{\sigma}^2)$, or Bernoulli distribution: $s_{u,i} \sim \mathbf{B}(\sigma(\phi(\boldsymbol{u}, \boldsymbol{i})))$, where $\sigma(\cdot)$ is the sigmoid function.

In Sec. 3.2.2, for a given user $u$, $\boldsymbol{s}_u \sim \text{Mult}(N_u, \boldsymbol{p}_u)$, where $\boldsymbol{p}_u = (p_{u1}, p_{u2}, ..., p_{uK})^T$ is a $K$-dimensional vector. The probability matrix $\boldsymbol{P}_{u2i}$ is defined with the same $p_{ui} = p(i|u)$. $p_{ui}$ is modeled as

$$p_{ui} = \frac{\exp \phi_\theta(\boldsymbol{u}, \boldsymbol{i})}{\sum_{\boldsymbol{j} \in \mathbb{I}} \exp \phi_\theta(\boldsymbol{u}, \boldsymbol{j})}. \tag{5}$$

In Sec. 3.3 of PUMS, we can define the specific $\boldsymbol{P}_{\text{MF}}$ and $\boldsymbol{P}_{i2u}$ in a similar manner. For example, we use $\boldsymbol{s}_i$ to denote the user who will interact with the item $i$, and $\boldsymbol{s}_i \sim \text{Mult}(N_i, \boldsymbol{p}_i)$, where $\boldsymbol{p}_i = (p_{1i}, p_{2i}, ..., p_{Mi})^T$ is a $M$-dimensional vector. $\boldsymbol{P}_{i2u}$ is defined with the same $p_{ui} = p(u|i)$, and it is modeled as

$$p_{ui} = \frac{\exp \phi_\theta(\boldsymbol{u}, \boldsymbol{i})}{\sum_{\boldsymbol{v} \in \mathbb{U}} \exp \phi_\theta(\boldsymbol{v}, \boldsymbol{i})}. \tag{6}$$

## A.2 LOSS FUNCTIONS

In this paper, we focus on Bernoulli and Multinomial distributions, because they usually lead to better results than Gaussian distribution (He et al. (2017); Rendle et al. (2020)). Following the maximum likelihood estimation (MLE) principle, the loss for Bernoulli distribution is:

$$l = -\frac{1}{|\mathcal{D}|} \sum_{u \in \mathbb{U}, i \in \mathbb{I}, (u,i) \in \mathcal{D}} s_{u,i} \log \sigma(\phi_\theta(u,i)) + (1 - s_{u,i}) \log(1 - \sigma(\phi_\theta(u,i))), \qquad (7)$$

where $s_{u,i} \in \{0,1\}$, and $s_{u,i} = 1$ are the positive entries in $\boldsymbol{S}_{ui}$, and $s_{u,i} = 0$ are negative samples randomly sampled from the entries in $\boldsymbol{S}_{ui}$, and $\mathcal{D}$ is the training dataset consist of all the positive and randomly sampled negative samples.

The loss for Multinomial distribution is softmax-cross-entropy loss (Liang et al., 2018):

$$l = \frac{1}{|\boldsymbol{S}_{u,i}|} \sum_{u \in \mathbb{U}, i \in \mathbb{I}, s_{u,i}=1} -\log \frac{\exp \phi_\theta(u,i)}{\sum_{i' \in \mathbb{I}} \exp \phi_\theta(u,i')}, \qquad (8)$$

where $|\boldsymbol{S}_{u,i}|$ is the sum of the entries in $\boldsymbol{S}_{ui}$.

In practice, the vocabulary $\mathbb{I}$ can be very large so that calculating the partition function $\sum_{i' \in \mathbb{I}} \exp \phi_\theta(u,i')$ becomes impractical. SSM and InfoNCE loss can be used to solve this problem:

$$l = \frac{1}{|\boldsymbol{S}_{u,i}|} \sum_{u \in \mathbb{U}, i \in \mathbb{I}, s_{u,i}=1} -\log \frac{\exp(\phi_\theta(u,i) - \log p_n(i|u))}{\exp(\phi_\theta(u,i) - \log p_n(i|u)) + \sum_{i' \in \mathbb{I}_u} \exp(\phi_\theta(u,i') - \log p_n(i'|u))}, \qquad (9)$$

where $\mathbb{I}_u \subset \mathbb{I}$ contains hundreds or thousands of negative samples, and $p_n(i|u)$ is a predefined background distribution. It has been shown that optimizing Eq. 9 will make $\phi_\theta(u,i)$ converge to $\log \hat{p}_{\text{data}}(i|u)$ (Jean et al., 2015; Oord et al., 2018).

## A.3 BI-INFONCE LOSS

Here we prove that $\phi_\theta(u,i)$ in Eq. 3 converges to $\log \hat{p}_{\text{data}}(u,i)$.

In Eq. 3, in the optimum of the first part, following the proof by Jean et al. (2015); Oord et al. (2018), we have $\frac{\exp(\phi_\theta(u,i))}{q_i} \propto \frac{\hat{p}_{\text{data}}(i|u)}{q_i}$, so we can assume:

$$\phi_\theta(u,i) = \log \hat{p}_{\text{data}}(i|u) + f(u) = \log \hat{p}_{\text{data}}(u,i) - \log \hat{p}_{\text{data}}(u) + f(u)$$

for a given $u$ and any $i$. Similarly, in the optimum of the second part, $\phi_\theta(u,i) = \log \hat{p}_{\text{data}}(u,i) - \log \hat{p}_{\text{data}}(i) + g(i)$ for a given $i$ and any $u$.

Thus, for any $u$ and $i$, this will always hold: $-\log \hat{p}_{\text{data}}(u) + f(u) \equiv -\log \hat{p}_{\text{data}}(i) + g(i)$, so it must be some constant. Then we have $\phi_\theta(u,i) = \log \hat{p}_{\text{data}}(u,i) + C$, where $C$ is some constant that is independent of $u$ and $i$. And when $\phi_\theta(u,i)$ converges to $\log \hat{p}_{\text{data}}(u,i)$, both parts in Eq. 3 can reach their optima.

## A.4 EQUIVALENCE BETWEEN BERNOULLI AND MULTINOMIAL DISTRIBUTIONS ON MODELING THE USER-ITEM INTERACTION MATRIX

Here we show that by properly choosing the training data distribution, $\phi_\theta(u,i)$ in Eq. 2 is actually approaching the conditional distribution $p(i|u)$ or $p(u|i)$, or joint distribution $p(u,i)$, or $\frac{p(u,i)}{p(u)p(i)}$.

Our derivation is inspired by Noise Contrastive Estimation (Gutmann & Hyvärinen, 2010).

In the modeling with Bernoulli distribution, we have two sets of samples, $\mathbb{X}$ and $\mathbb{Y}$. $\mathbb{X}$ is the set of positive samples and $\mathbb{Y}$ is the set of negative samples randomly sampled based on a certain distribution $p_n(\cdot)$.

Assume $\mathbb{X} = \{\boldsymbol{x}_1, \boldsymbol{x}_2, ..., \boldsymbol{x}_L\}$ contains $L$ samples, and $\mathbb{Y} = \{\boldsymbol{y}_1, \boldsymbol{y}_2, ..., \boldsymbol{y}_T\}$ contains $T$ samples, and $\mathbb{Z} = \mathbb{X} \cup \mathbb{Y} = \{\boldsymbol{z}_1, \boldsymbol{z}_2, ..., \boldsymbol{z}_{L+T}\}$ contains all the $L+T$ samples. Here $\boldsymbol{x}_l := (u,i)$, $\boldsymbol{y}_t := (u,i)$ and $\boldsymbol{z}_j := (u,i)$, where $u \in \mathbb{U}$ and $i \in \mathbb{I}$. We assign each sample $\boldsymbol{z}_j$ a binary class $C_j$: $C_j = 1$ if $\boldsymbol{z}_j$ comes from $\mathbb{X}$, and $C_j = 0$ if $\boldsymbol{z}_j$ is from $\mathbb{Y}$.

We assume the joint probability of positive samples in $\mathbb{X}$ is parameterized by $\theta$ is $p_{\mathrm{model}}(u, i; \theta) = p_{\mathrm{model}}(\boldsymbol{z}; \theta)$. So we have the conditional probabilities:

$$p(\boldsymbol{z}|C = 1; \theta) = p_{\mathrm{model}}(\boldsymbol{z}; \theta) \qquad p(\boldsymbol{z}|C = 0; \theta) = p_n(\boldsymbol{z}). \tag{10}$$

The posterior probabilities are:

$$p(C = 1|\boldsymbol{z}; \theta) = \frac{p_{\mathrm{model}}(\boldsymbol{z}; \theta)P(C = 1)}{p_{\mathrm{model}}(\boldsymbol{z}; \theta)P(C = 1) + p_n(\boldsymbol{z})P(C = 0)} = \frac{1}{1 + \exp(-G(\boldsymbol{z}; \theta))}, \tag{11}$$

where

$$G(\boldsymbol{z}; \theta) = \log \frac{p_{\mathrm{model}}(\boldsymbol{z}; \theta)P(C = 1)}{p_n(\boldsymbol{z})P(C = 0)}, \tag{12}$$

and we also have

$$p(C = 0|\boldsymbol{z}; \theta) = \frac{1}{1 + \exp(G(\boldsymbol{z}; \theta))}.$$

So the log-likelihood is:

$$l(\theta) = \sum_{j=1}^{L+T} C_j \log P(C_j = 1|\boldsymbol{z}_j; \theta) + (1 - C_j) \log P(C_j = 0|\boldsymbol{z}_j; \theta).$$

Through optimizing the binary classification using samples in $\mathbb{Z}$ and the corresponding binary classes $C$, we are actually recovering the MF with Bernoulli distribution defined in Sec. 3.2.1. The dot product $\phi_\theta(u, i)$ will converge to $G(\boldsymbol{z}; \theta)$ in Eq. 12:

$$\phi_\theta(u, i) \to \log \frac{p_{\mathrm{model}}(\boldsymbol{z}; \theta)P(C = 1)}{p_n(\boldsymbol{z})P(C = 0)}. \tag{13}$$

Different negative sampling strategies will lead to very different optimal $\phi_\theta(u, i)$. For example, if we randomly sample $n$ items for each positive $(u, i)$ pair to form negative samples with the $u$, then we have $p_n(\boldsymbol{z}) = p_n(u, i) = \hat{p}_{\mathrm{data}}(u) \cdot 1/K$, where $\hat{p}_{\mathrm{data}}(u)$ is the empirical marginal probability of the positive samples. Substitute $p_n(\boldsymbol{z})$ into Eq. 13, we have

$$\phi_\theta(u, i) \approx \log \frac{p_{\mathrm{model}}(u, i; \theta)}{\hat{p}_{\mathrm{data}}(u)} + C \approx \log p_{\mathrm{model}}(i|u; \theta) + C.$$

Similarly, if we randomly sample $n$ users for each positive $(u, i)$ pair to form negative samples with the $i$, then $\phi_\theta(u, i)$ will converge to the conditional probability $\log p_{\mathrm{model}}(u|i; \theta)$.

For each positive $(u, i)$ pair, if we randomly sample $n$ items to form negative samples with the $u$, and also $n$ users to form negative samples with the $i$, then $\phi_\theta(u, i)$ will converge to the point-wise mutual information $\log \frac{p_{\mathrm{model}}(u, i; \theta)}{p_{\mathrm{model}}(u; \theta)p_{\mathrm{model}}(i; \theta)}$.

And if the negative samples are randomly sampled uniformly from all users and items globally, then $\phi_\theta(u, i)$ will converge to the joint probability $\log p_{\mathrm{model}}(u, i; \theta)$.

Therefore, we show that MF with Bernoulli distribution and Multinomial distribution can be equivalent if we choose the proper negative sampling methods.

## A.5 MOTIVATION AND PRACTICAL APPLICATIONS OF TR

For the example of mining potential users given 1 billion users and 10,000 items, we list all the alternative methods that we can think of:

1. For each item, we create the training sample, and then train a binary classification model, and finally predict for the 1 billion users and pick the top 1 million. We need to repeat 10,000 times.

2. By emulating the RS, we model $p(u|i)$ and obtain the embeddings of users and items, and then apply the ANN search (e.g., Faiss) to find the top 1 million users for each item. This method has a serious defect: ANN has high accuracy if we retrieve the top hundreds or thousands of candidates, but its accuracy and efficiency drops dramatically if we extend the range to millions. So, this method is not practical.

3. Also, we model $p(u|i)$ and obtain the embeddings of users and items, and then do a brute force search: we calculate the similarity score of all the users and items, which is the Cartesian product of 1 billion users and 10,000 items, i.e., 10 trillion, and then we pick the top 1 million users for each item. It's very inefficient and consume huge number of resources.

4. We build two models for $p(i|u)$ and $p(u|i)$ respectively. Then we use the embeddings of $p(i|u)$ and apply the ANN search to find the top 10 items for each user. Now we have 10 billion u-i pairs, and we use the embeddings from $p(u|i)$ to calculate the similarity score of the u-i pairs, and finally pick the top 1 million for each item.

TR has just one less model than the last alternative and one less step, but in practice, it is problematic to maintain two correlated models, especially when they are very large. Two models usually imply that the calculation is doubled, the storage is doubled, and the risk and instability are doubled.

To conclude, TR is the best choice that can balance the accuracy, efficiency, risk and maintainability in practice.

## A.6 EXPERIMENT RESULTS: RS

### A.6.1 MF WITH BERNOULLI DISTRIBUTION

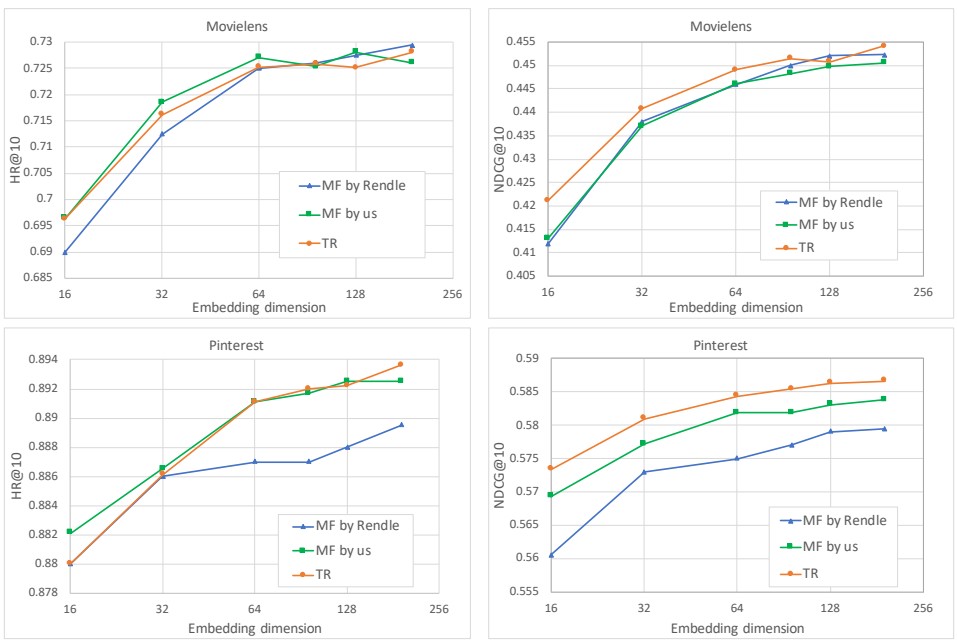

Figure 2: Comparison of results of MF given by Bernoulli distribution and Multinomial distribution. 'MF by Rendle' is from Rendle et al. (2020), 'MF by us' is our implementation. In general these results are quite close and outperform Neural Collaborative Filtering by a large margin (Rendle et al. (2020)).

We compare to MF in Rendle et al. (2020) using the same datasets: binarized Movielens-1m (Harper & Konstan, 2015) and a dataset from Pinterest (Geng et al., 2015).

**Train and Evaluation Configurations**

For each user, the last item is held out as the test set, and the second last is held out as the validation set, and the remaining as the training set. We use the validation set and training set to select the best hyper-parameters, and then re-train the model using both training and validation sets.

During evaluation, each user has his own candidates pool, and it consists of the withheld item (i.e., the ground-truth) and 99 sampled negative items. These methods will score the 100 items and rank them by the score.

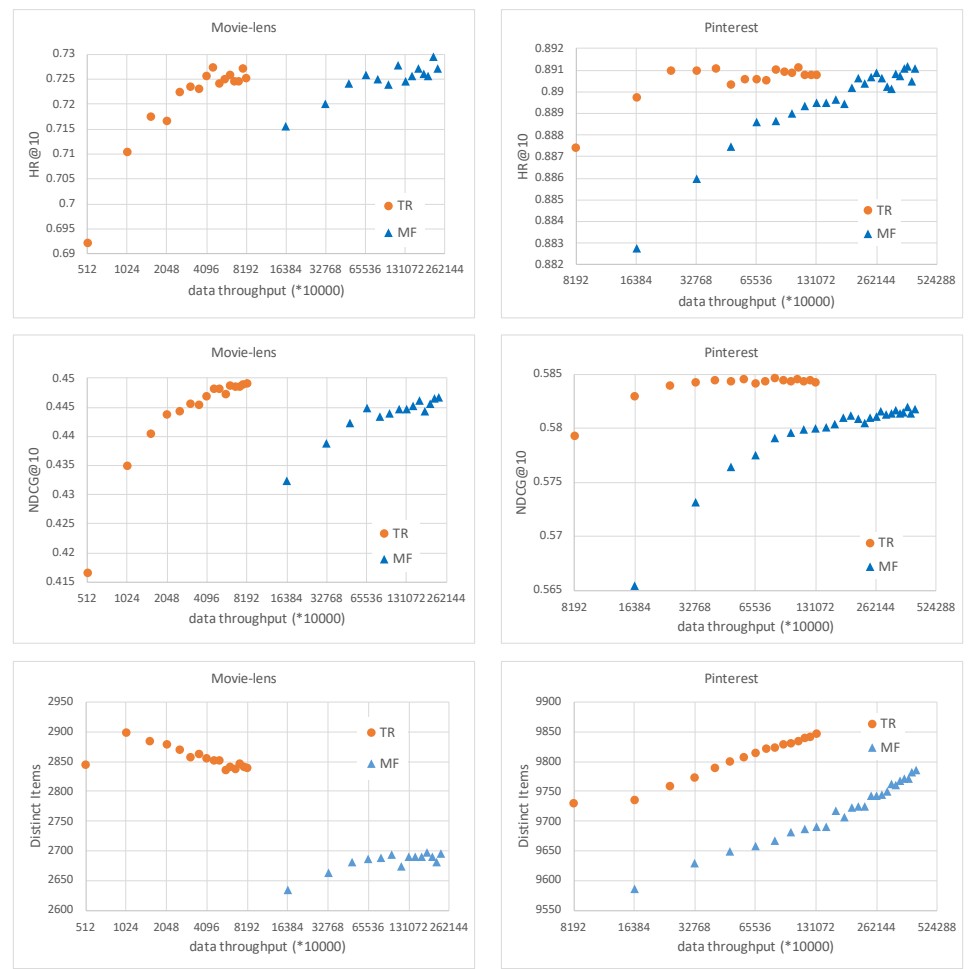

Figure 3: Comparison of the converging speed and diversity of MF modeled with Bernoulli distribution (MF) and Multinomial distribution (TR). The $x$-axis is the data throughput, i.e., the number of records consumed by the model, and the $y$-axis is the metric measured on the test data. For Movie-lens, TR is about $16\times$ times faster, and for Pinterest it is about $10\times$ times.

The hyper-parameters $\tau$ and batch-size are chosen using the validation data. For Movielens-1m, we use $\tau = 0.125$, batch-size $= 4096$ for $\text{MF}_b$, and $\tau = 0.111$, batch-size $= 512$ for TR. For Pinterest, we use $\tau = 0.143$, batch-size $= 8192$ for $\text{MF}_b$, and $\tau = 0.167$, batch-size $= 8192$ for TR. We have $d \in \{16, 32, 64, 96, 128, 196\}$ for a comprehensive comparison.

**Evaluation Metrics**

Two evaluation metrics are employed to measure the performance of top-$N$ ranked items:

$$\text{HitRate@}N = \frac{1}{|\mathbb{U}|} \sum_{u \in \mathbb{U}} \delta(|\hat{\mathcal{I}}_{u,N} \cap \mathcal{I}_u| > 0), \tag{14}$$

where $\hat{\mathcal{I}}_{u,N}$ denotes the top-$N$ ranked items for $u$, $\mathcal{I}_u$ is the set of testing items, and $\delta(\cdot)$ is the indicator function.

$$\text{NDCG@}N = \frac{1}{|\mathbb{U}|} \sum_{u \in \mathbb{U}} \frac{1}{\log(1 + r_u)}, \tag{15}$$

where $r_u$ is the rank of the withheld item in the top-$N$ items for $u$.

**Results**

The performance of different methods is shown in Fig. 2. In general, MF with Bernoulli distribution and Multinomial distribution are comparable, which is consistent with our theory in Sec. 3.5.

The comparison of the converging speeds is shown in Fig. 3. We can see that modeling with Multinomial distribution is much better than Bernoulli distribution.

### A.6.2 SEQUENTIAL MODELING WITH MULTINOMIAL DISTRIBUTION

We compare to the models implemented in ComiRec (Cen et al., 2020). ComiRec provides the codes and two preprocessed datasets: Amazon books data (Ni et al., 2019) and Taobao click data (Zhu et al., 2018). So we use the same datasets and evaluation metrics for a fair comparison.

**Metrics and Comparing Methods**

We use HitRate@$N$ and Recall@$N$ to measure the accuracy. HitRate@$N$ is defined in Eq. 14, and Recall@$N$ is defined below:

$$\text{Recall@}N = \frac{1}{|\mathbb{U}|} \sum_{u \in \mathbb{U}} \frac{|\hat{\mathcal{I}}_{u,N} \cap \mathcal{I}_u|}{|\mathcal{I}_u|}. \tag{16}$$

Metrics 'Distinct items@$N$' counts the distinct items of all users' top-$N$ recommended list, and 'Distinct cates@$N$' counts the distinct categories of these items. They measure the diversity of the recommendation results.

The comparing methods include the following:

- **MostPopular** recommends the most popular items to all users.
- **YouTube DNN** (Covington et al., 2016) is a simple and very efficient neural network model adopted in the CG stage of the large-scale industrial RS.
- **ComiRec-SA & ComiRec-DR** (Cen et al., 2020) use self-attention (Vaswani et al., 2017) or dynamic routing (Sabour et al., 2017) to explicitly incorporate users' diverse interests in the model's architecture to increase accuracy and diversity.

**Results**

The results evaluated by the above metrics can be found in Fig. 4. We can see that their HitRate and Recall are comparable, and this is consistent with our theory in Sec. 3.5. The diversity of TR with Bi-InfoNCE is much better than other methods.

The performance of next-$n$-prediction are shown in Fig. 5. We investigate the KL Divergence of the marginal distributions of train and test data in Fig. 6, and show that the performance gain in Amazon data may be due to the reduced discrepancy between the train and test data.

In Taobao data, the discrepancy gets larger when we increase $n$, and in experiments the accuracy does not increase.

### A.7 EXPERIMENT RESULTS: PUMS

There are different ways of creating the validation and test datasets, such as randomly picking 10 users for each chosen item or each group of items.

Here we choose to holdout 1% users for the practical reason: usually we mine the potential users based on the historical behaviors and then deliver advertisement to them in the future, and the implicit assumption is that the distribution of the future is close to the past, i.e., popular items in the

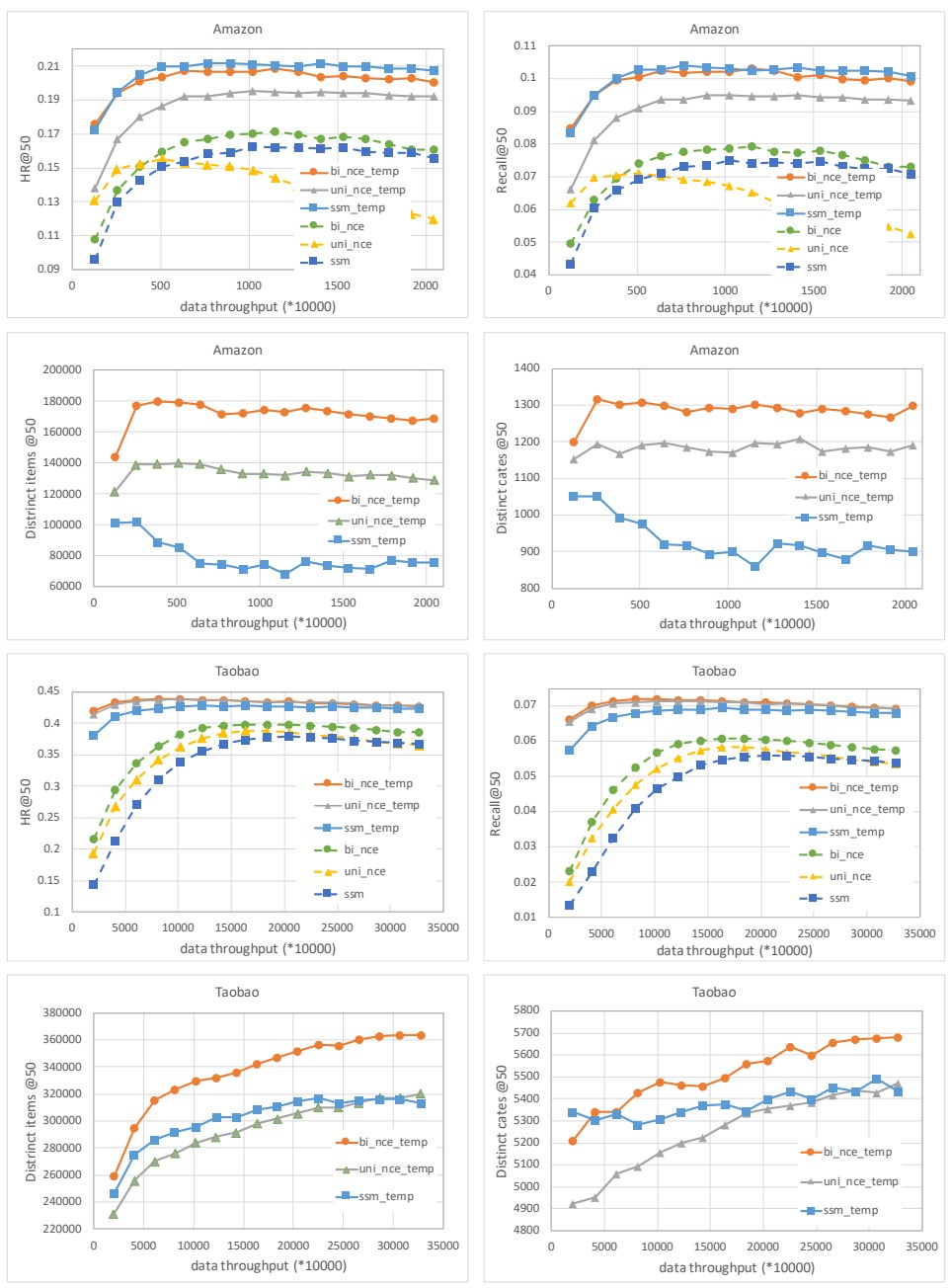

Figure 4: Comparison of accuracy and diversity metrics of different methods on Amazon books and Taobao data. HitRate@50 and Recall@50 measure the accuracy, and Distinct items@50 and Distinct cates@50 measure the diversity. '∗_temp' means the users and items representations $u$ and $i$ are l2-normalized, and then their dot product is rescaled with temperature $\tau = 0.067$. By this way, the model converges much faster to higher accuracy. As for diversity, Bi-InfoNCE surpass Uni-InfoNCE and SSM by a large margin.

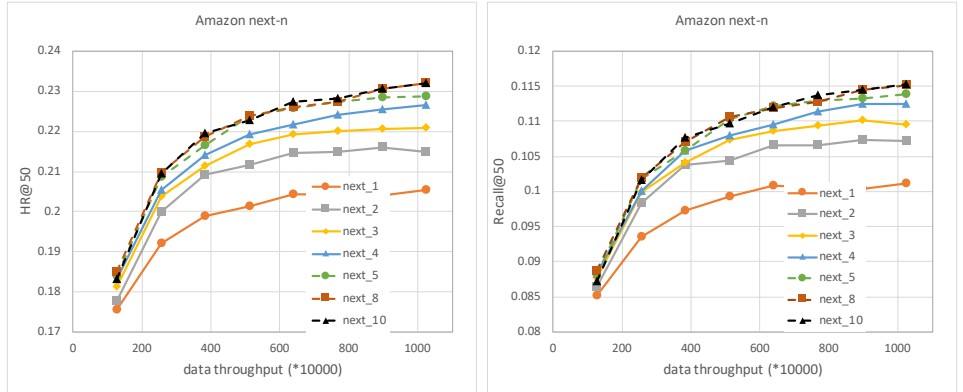

Figure 5: Results of Next-$n$-prediction for Amazon data. The improvement of accuracy is substantial when $n <= 5$.

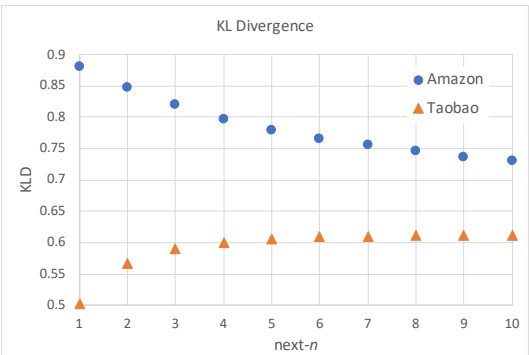

Figure 6: KL Divergence between two marginal distributions of items: $\hat{p}_{\text{test}}(i)$ of test data and $\hat{p}_{\text{data}}(i)$ of next-$n$ training data. Although they are not the full distributions of $u$ and $i$, to a certain extent they can reveal the discrepancy between the distributions of the training data and the test data. In Amazon data, if we increase $n$, the discrepancy becomes smaller, and the accuracy also increases (See Fig. 5). In Taobao data, the discrepancy gets larger when we increase $n$, and in experiments the accuracy does not increase.

Table 4: Statistics of the split Movielens datasets in PUMS experiments.

|  | # records | # movies | # users |
|---|---|---|---|
| train (w/o negative samples) | 990296 | 3076 | 6040 |
| valid | 5125 | 1257 | 2698 |
| test | 4788 | 1257 | 2547 |

past will be bought by more users in the future, and vice versa. So we holdout validation and test datasets in accordance with the overall distribution to emulate the practical scenario.

### A.7.1  POTENTIAL USERS OF A SINGLE ITEM

We create the training, validation and test data as described in Sec. 5.2. The statistics of the resulting datasets is shown in Tab. 4.

**The Metrics**

Our evaluation metrics of accuracy are Precision@$N$ and Recall@$N$:

$$\text{Precision@}N = \frac{1}{\mathbb{I}_{\text{test}}} \sum_{i \in \mathbb{I}_{\text{test}}} \frac{|\hat{\mathcal{U}}_{i,N} \cap \mathcal{U}_i|}{N},$$

where $\hat{\mathcal{U}}_{i,N}$ denotes the top-$N$ ranked users for $i$ and $\mathcal{U}_i$ is the set of users for the testing item $i$.

$$\text{Recall@}N = \frac{1}{\mathbb{I}_{\text{test}}} \sum_{i \in \mathbb{I}_{\text{test}}} \frac{|\hat{\mathcal{U}}_{i,N} \cap \mathcal{U}_i|}{|\mathcal{U}_i|}.$$

We count the distinct users of all the test items to measure the diversity.

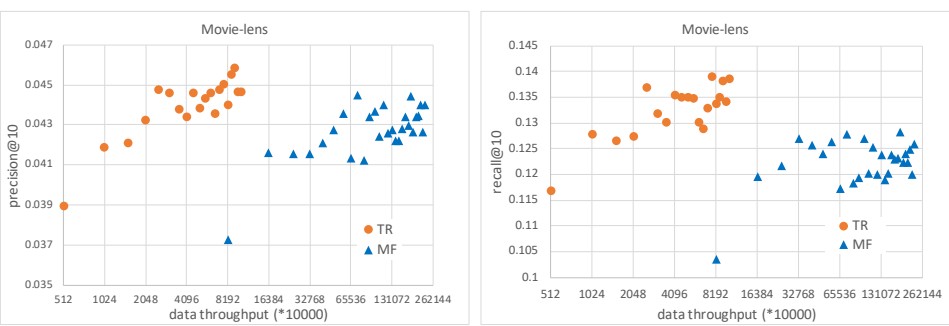

Figure 7: Comparison of the converging speed between MF and TR. The $x$-axis is in log scale. We can see that TR can obtain better results much faster.

**Results of Converging Speed**

The comparison of the converging speed can be found in Fig. 7. Similar to the results in Fig. 3 of RS, TR is much faster than MF with Bernoulli distribution.

### A.7.2  POTENTIAL USERS OF A GROUP OF ITEMS

For a given genre, we define the popularity of its movies in this way: we first rank the movies based on the number of users who have watched them, and then we pick the top/middle/bottom 10% movies as different groups. So we obtained 54 groups from 18 genres.

In practice, movies of different genres and different popularity will behave very differently, e.g., precision and recall of popular Action movies and unpopular Western movies will differ a lot. Movie-

Table 5: Overall performance of mining the potential users of a group of items. The % is omitted for precision and recall. The hyper-parameters of $MF_b$ are batch-size $= 4096$ and $\tau = 0.125$, and for TR batch-size $= 512$ and $\tau = 0.111$. We use $d = 96$ for all models.

| model | precision@10 | recall@10 | distinct users |
|---|---|---|---|
| $MF_b$ [0:8] *mean* | 5.93 | 1.52 | - |
| $MF_b$ [8:0] *sum* | 5.93 | 2.58 | - |
| $MF_b$ [8:0] *max* | 14.63 | 4.23 | - |
| $MF_b$ [8:0] *mean* | 17.04 | 4.57 | 246 |
| $MF_b$ [4:4] *mean* | 11.11 | 2.63 | - |
| $MF_b$ [8:8] *mean* | 12.78 | 3.05 | - |
| TR [Uni-InfoNCE $u2i$] *mean* | 4.63 | 1.38 | - |
| TR [Uni-InfoNCE $i2u$] *mean* | 16.85 | 7.25 | **262** |
| TR [Bi-InfoNCE] *sum* | 12.41 | 6.79 | - |
| TR [Bi-InfoNCE] *max* | 16.48 | 7.01 | - |
| TR [Bi-InfoNCE] *mean* | **20.56** | **7.58** | 258 |

lens have 18 genres, and we divide the movies of the same genre into 3 categories based on their popularity, so we can split the chosen movies into 54 groups.

Given a group and its affined items, we define its potential users this way: if a user interacts with one or more items in the group (excluding the items he has interacted previously), then he will be the potential users of the group. For example, given the group $\mathbb{G} = \{i_1, i_2, i_3\}$ and the user $u_1$, if $u_1$ has interacted with $i_1$ in the historical data, then we cannot pick him as the potential user; if in the future data, $u_1$ interact with $i_2$ or $i_3$, then he will be regarded as the potential user of $\mathbb{G}$.

Different groups have different number of movies, e.g., the group 'Drama-top10%' contains 93 movies, while 'Documentary top10%' has only 3 movies, and their precision and recall will also differ a lot. We report the averaged precision and recall over all groups in Tab. 5.

For small datasets like movie-lens-1m, the model generates the potential users of a group this way: for each item in the group, we first calculate $\phi_\theta(\boldsymbol{u}, \boldsymbol{i})$ of Eq. 2, and then the score $\hat{p}_{TR}(\boldsymbol{u}, \boldsymbol{i}) = \exp(\phi_\theta(\boldsymbol{u}, \boldsymbol{i}))$ for TR and $\hat{p}_{MF}(\boldsymbol{u}, \boldsymbol{i}) = \sigma(\phi_\theta(\boldsymbol{u}, \boldsymbol{i}))^4$ for $MF_b$, with respect to the candidate users. Thus each candidate user will have several scores, and then we aggregate the scores to obtain a match score between the user and the group, and finally pick the top-$k$ users. The aggregation methods could be *max*, *mean* or *sum*.

In Tab. 5, '$MF_b$ [0:8]' and 'TR [Uni-InfoNCE $u2i$]' are specifically for $u2i$ CG of RS, so they give the worst results here. Models tailored for $i2u$ ('$MF_b$ [8:0]' and 'TR [Uni-InfoNCE $i2u$]') and the general model for bidirectional CG ('TR [Bi-InfoNCE]') perform much better.

The aggregation methods also affect the accuracy, and *mean > max > sum*. If a user has scores with many items of the group, the *sum* could be high even if individual scores are low, and this also implies that the user has few interactions with the items of the group in the historical data. So it would not be suitable that *sum* ranks this user high. In contrast, *mean* and *max* will not have this problem, and *mean* considers more scores than just the *max* one and gives best results.

---

[4] $\sigma(\cdot)$ is the sigmoid function.

