# OpenReview forum: "TotalRecall: A Bidirectional Candidates Generation Framework for Large Scale Recommender \& Advertising Systems"
_ICLR.cc/2022/Conference — ICLR 2022 Submitted_

### Official Review · Reviewer_qfRw · 2021-10-27

**Correctness:** 3
**Technical Novelty And Significance:** 2
**Empirical Novelty And Significance:** 2
**Recommendation:** 3
**Confidence:** 4

**Main Review:**

The main strengths of this work are listed as follows:

1.	The proposed Bi-InfoNCE loss is easy to deploy.

2.	TR has improvements on both accuracy and diversity.

However, this work has the following weaknesses:

1.	It seems that the main contribution of this work is the proposed Bi-InfoNCE loss. However, similar contrastive learning (Uni-InfoNCE) method [1] has been proposed and verified in candidate generation module. It is straightforward that both users and items could be used as negative samples if we attempt to jointly provide results for recommender system (which provides items for a user) and advertising system (which provides users for an item). Therefore, the technical novelty is limited.

2.	Related recommendation models [1] should be compared.

3.	In PUMS, the authors only conduct evaluations on MovieLens-1M, which is not a real widely-used advertising dataset. Moreover, the baseline MF is also not sufficient and competitive enough in advertising. I suggest that the authors should conduct PUMS evaluations on real advertising datasets with SOTA advertising models.

4.	The universality of TR is not verified. The authors should deploy TR on different real-world recommendation and advertising models for comprehensive comparisons.

5.	In Table 2, why not use the original next-item-prediction task in evaluation?

6.	In Table 3, will more negative users help to improve the performance of MF (e.g., MF_b[16:0]).

7.	In Eq. (3), The Bi-InfoNCE loss further considers negative users compared to the Uni-InfoNCE loss used in [1]. However, does the improvements of Bi-InfoNCE mainly derive from more negative samples? Since larger batch sizes may improve the performance given in [2]. The authors can select two negative user sets in Eq. (3), or just double the batch size of the Uni-InfoNCE loss for more comparisons.


References:

[1] Zhou C, Ma J, Zhang J, et al. Contrastive learning for debiased candidate generation in large-scale recommender systems[C]//Proceedings of the 27th ACM SIGKDD Conference on Knowledge Discovery & Data Mining. 2021: 3985-3995.

[2] Chen T, Kornblith S, Norouzi M, et al. A simple framework for contrastive learning of visual representations[C]//International conference on machine learning. PMLR, 2020: 1597-1607.


**Summary Of The Paper:**

This work proposes a bidirectional candidate generation framework for both recommender system (which provides items for a user) and advertising system (which provides users for an item). Specifically, the authors conduct a Bi-InfoNCE loss on the classical two-tower architecture to jointly learn both user and item representations. The Bi-InfoNCE loss considers each (u, i) pair as a positive instance, and selects both users and items as negative instances in InfoNCE. In experiments, the authors find that the proposed TR model outperforms MF in both recommender system and advertising system datasets.

**Summary Of The Review:**

The technical novelty is limited. Moreover, the improvements and universality of TR are not perfectly verified.

In conclusion, I will vote for Reject.

---

> ### Author Response · Authors · 2021-11-22
> **Reply to Reviewer qfRw**
>
> Thank you very much for your comments and suggestions. Our reply is as follows:
>
> 1.	Technical contribution
>
> A: First, our work differs from CLRec as follows:
>
> i). Our work is the concurrent work of CLRec, and we were not aware of CLRec when we were developing TR.
>
> ii). CLRec's modeling objective is the point-wise mutual information p(i|u)/p(i), because it doesn't subtract the logarithm of the background distribution p(i). This objective might be helpful in so-called debiasing scenario, but once evaluated with the usual u2i metrics like HR and Recall, the performance is much worse so that we didn't include it in the paper. In contrast, our Uni-InfoNCE models p(i|u).
>
> iii). The development of TR is deeply rooted in our practical applications. (See our reply "Reply to two common concerns: Motivation and Technical Contribution" for details.) The Bi-directional modeling allow us to efficiently do PUMS and RS. And the proof of the modeling objective of joint probability distribution p(u,i) supports our applications in theory.
>
> Lastly, we have made a new technical contribution as shown in our reply "A new theoretic discovery".
>
> 2.	Compare to CLRec
>
> A: As explained in 1, we didn't compare with CLRec.
>
> 3.	Comparison in PUMS
>
> A: Actually, we didn't find many literatures studying the i2u candidate generation. As discussed in our "Related Works", practitioners usually use binary classification algorithms to model the i2u for each i separately. Also, they may use MF_b[8:0] to model the i2u for all the items together, but with more complicated architectures (e.g., Deep Interest Network's architecture.)
>
> Here we want to ensure a fair comparison, so we fix the architecture to be the same for different methods, i.e., MF_b[?:?], Uni-InfoNCE and Bi-InfoNCE.
>
> Also, you may refer to our discussion of the motivations in our reply "Reply to two common concerns: Motivation and Technical Contribution".
>
> 4.	Universality of TR
>
> A: We apologize that we might not state clearly about the motivations and applications in the previous draft. As discussed in our reply "Reply to two common concerns: Motivation and Technical Contribution", we develop TR from the practical point of view, and it can balance the accuracy and efficiency in practice.
>
> In the paper, we show that our method has the solid theoretical basis, and can archive comparable or even better performance compared to the SOTA modeling objectives given the same model architecture and capacity.
>
> 5.	Table 2, next-item-prediction
>
> A: In Table 2, only the last evaluation uses next-7-item-prediction. Others use next-item-prediction.
>
> 6.	Table 3, more negatives, e.g., MF_b[16:0]
>
> A: We tried but it didn't work. Besides, as pointed out in our reply "A new theoretic discovery", MF_b[16:0] and MF_b[8:0] are equivalent in terms of the modeling objective. They both make the \phi(u,i) converging to log{p(u|i)}.
>
> 7.	Bi-InfoNCE vs Uni-InfoNCE
>
> A: Thanks for diving deep into the details. We think that more negatives will let Bi-InfoNCE converges faster to the optimum compared to Uni-InfoNCE, as observed in the experiments.
>
> The performance gain of Bi-InfoNCE in terms of the metrics of u2i candidate generation is subtle. Theoretically, modeling p(u,i) and p(i|u) will not make much difference if evaluated with the metrics designed for p(i|u). Results of p(u,i) will be much better if they [p(u,i) and p(i|u)] are evaluated with the metrics designed for p(u|i).
>
>
> Thank you again for your comments. We hope that our reply addresses your concerns. We will update our draft accordingly.

---

### Official Review · Reviewer_knEH · 2021-10-29

**Correctness:** 2
**Technical Novelty And Significance:** 2
**Empirical Novelty And Significance:** 2
**Recommendation:** 3
**Confidence:** 4

**Main Review:**

Pros:

The overall architecture seems to be novel in the literature to my knowledge. This paper has tried to evaluate the proposed method from various aspects through several experiments.

Cons:

One of the claims will be that jointly addressing recommendations and advertising is effective in e-commerce services. However, it seems this paper did not sufficiently motivate the claim. Although Sec. 1 and Sec. 2.2 cite several related studies, those studies lack a discussion of situations in which both recommendation and advertising are performed in one e-commerce platform. I think it would help the reader's understanding if this paper explains how such platforms do recommendations and advertising and how the proposed joint approach is helpful for them in more detail. Also, if any previous studies perform both recommendation and advertising simultaneously, I recommend citing them.

While the proposed method will contain some promising ideas, it seems that its technical novelty is somewhat weak. For example, SimCLR (Chen et al., 2020) has considered the idea of using the l2 normalization and temperature parameter of a similar score function. The Bi-InfoNCE loss is an appealing idea but does not seem to be completely novel. Similar ideas have already been proposed in different contexts [1,2,3]. Since those papers have recently been made available, I perceive them as contemporaneous work of this paper. However, considering them, I feel the derivation of the Bi-InfoNCE loss is reasonable but fairly straightforward, and the uniqueness and technical contribution is somewhat weak.

Moreover, this paper did not explain well how the proposed method is effective in Sec. 3. For example, this paper calls the proposed method 'TotalRecall.' I interpreted the proposed method takes care of the improvement of recall, but Sec. 3 did not provide any implications. Also, this paper claims that the proposed method was effective in terms of convergence speed and diversity in the experiments. However, Sec. 3 did not clearly explain what part of the proposed method induces such effectiveness. I feel that the contribution of the proposed method is somewhat weak if it just happens to perform well in the experiments. For example, in Sec. 3.2.2, this paper claims modeling s_u can 'boost the converging speed a lot.' I recommended presenting the reason more quantitatively in Sec. 3 rather than just showing experimental results.

While the experiments tried to evaluate the proposed method from various perspectives, the results do not always strongly support the effectiveness of the proposed method. For example, the performance of the Uni-InfoNCE outperformed the one of the Bi-InfoNCE in Table 2. In Fig. 4, the 'bi_nce_temp' method does not significantly improve the results compared to the 'ssm_temp' method for HR and recall. Also, there are many parts where noteworthy experimental results are missing. In Sec. 4.1, the experiments did not show the results for Uni-InfoNCE. There are two variants of Uni-InfoNCE (u2i and i2u), but the results of i2u are not shown in Table 2. Although many missing results may not work out so well, I recommend presenting more comprehensive and consistent results for all experiments.

The clarity of the paper appears to be somewhat poor. This paper has so many unclear parts about the proposed method and experiments. Please see the comments below.

- The notation 'N' is used many times with different meanings in the whole paper. I recommend consistently using it throughout the manuscript.
- In the experiments, how many negative samples were used for all the methods in each experiment? I interpreted this paper used 100 negative samples in Sec. 4 according to the description of A.4.1. Did the TR methods also use 100 samples in Sec. 5? If so, it means that the TR methods used much larger samples than MF_b methods. Why?
- In the experiments, this paper refers too much to the figures in the appendix to explain the main experimental results. The appendix would be a material to support the main part, and I think it would be better to put the main results in the appendix much more sparingly.
- In the experiments, why was the effect of different sizes of negative samples only evaluated in Sec. 5? It would be more consistent to do the same assessment in Sec. 4.
- In the experiments, please ensure what the bold and underline highlighting means for all tables. Table 4 appears to have duplicate underlines.
- Sec. 1: this paper mentions 'Theoretically we show that the superiority is due to the modeling of joint probability of u and i.' However, I did not find any theoretical claims to show the superiority of the joint distribution.
- Sec. 2.3: I recommend specifying x and y of Eq. (1).
- Sec. 3.1: what exactly is y_{u,t}? Is this an indicator variable of a specific item that was clicked on by u at time t?
- Sec. 3.2.2 and A.1: I did not know what the variable s_u indicates. According to the text, s_u is drawn from the multinomial distribution Mult(N_u, p_u), which means that s_u is a K-dimensional count vector where s_{u, i} (i=1, ..., K) denotes the i-th element of s_u and corresponds to the number of interactions between u and i. However, Eq. (5) refers to s_u as an indicator of item i. Also, if S_{u, i} is a binary matrix, s_u will result in a binary vector. As in (Liang et al., 2018), the observation of s_u may happen to be binary, but this paper fixes the variable to be binary in the problem definition of Sec. 3.1.
- Sec. 3.4.1: I did not understand why this paper chose a classical two-tower architecture. It seems that the presented two reasons do not exclude other possible architectures.
- Sec. 3.4.1: 'in online serving .' -> 'in online serving.'
- Sec. 3.4.1: the notation of f_{\theta}(u) is confusing since it quite overlaps the notation of f_{\theta}(u, i) in Eq. (1). Also, f_{\theta}, g_{\theta}, and \phi_{\theta} are also confusing since they share the same parameter \theta. I recommend using different variables or specifying the detail of \theta.
- Sec. 3.4.1: \phi_{\theta} is used for both unnormalized and normalized score functions, which are confusing a little. For both functions, it is not clear what \theta is.
- Sec. 3.4.2: the definition of |S_{u, i}| seems unclear in Eq. (3). S_{u, i} is defined as a matrix, and the notation of |S_{u, i}| is not specified.
- Sec. 4.1: 'NCF' of the caption of Figure 2 is not defined in this paper.
- Sec. 4.1 and Sec. 5.2: this paper claims the TR methods converge faster than compared methods from the results of Fig. 3 and Fig. 7, but their x-axis denotes the number of processed data samples. I recommend showing the results with actual runtime on the x-axis.
- Sec. 4.1: it would be better to explain what distinct items are before introducing Fig. 3. The term 'distinct items' is first explained in A.4.2.
- Sec. 4.2: I didn't understand how to make the data for next-n-item prediction, which should be explained in Sec. 3.1. Also, this paper should ensure that using such data leads to fair comparison for all methods.
- Sec. 4.2: why did not the experiment report the NDCG as in (Cen et al., 2020)?
- Sec. 4.2: the difference of four (TR-)Youtube methods in Table 2 seems to be not explained.
- Sec. 4.2: why did not Table 2 report the result of 'TR-Youtube [next-7-prediction]' for the Taobao dataset?
- Sec. 4.2: how does each method in Fig. 4 correspond to the method in Table 2? Also, I did not fully understand the meaning of '_temp' in the caption of Fig. 4.
- Sec. 4.2: what is the coherent vector? How is it induced by Eq. (3) and how does it improve diversity?
- Sec. 5: in the first paragraph, please provide more specific examples that motivate mining for a group of similar items in real E-commerce services.
- Sec. 5: in the first paragraph, this paper mentions 'Previously, ... so many models.' However, in the experiment of Sec. 5, only 54 item groups were used. I think binary classification approaches are also applicable for item groups.
- Sec. 5.1: this paper mentions that 'We use Movielens-1m because it is also used in MFb of RS.' However, the Pinterest dataset was also used in the experiment of Sec. 4. Why was only the Movielens-1m dataset used?
- Sec. 5.2: this paper mentions 'TR uses the same training data and model configurations for both RS and PUMS.' However, the procedure for creating the datasets appears to be different in Sec. 4 and Sec. 5. I didn't know if this paper used the same TR models for both experiments.
- Sec. 5.2 and Sec. 5.3: I recommend showing all the scores of the distinct users even if the accuracies are very low.
- Sec. 5.3: 'For small dataset' -> 'For small datasets'?
- Sec. 5.3: I recommend showing all the results of aggregation methods: several results with max and sum are missing in Table 4.
- A.1: did s_{u, i} follow the Gauss distribution even if it is a binary random variable?
- A.2: S -> S_{u,i}?
- A.2: what is exactly N in Eq. (8)? Intuitively, I think Eq. (8) should be divided by M * K.
- A.3: I did not understand the claim of this section. I recommend providing more formal proofs to show the convergence of the score function.
- A.4: I think this section is not well-structured. Although the titles of subsections indicate the considered methods, they explain the evaluation metrics.


[1] Nguyen et al., Contrastive Learning for Natural Language-Based Vehicle Retrieval, CVPR 2021 Workshop.

[2] Bai et al., Connecting Language and Vision for Natural Language-Based Vehicle Retrieval, CVPR 2021 Workshop.

[3] Lee et al., Compressive Visual Representations, arXiv:2109.12909.

==========

I appreciate that the authors have responded to my concerns one by one. While some of the responses are convincing and have been reflected in the manuscript, it appears that not all the responses have been considered in the revision yet. I would like to see the completed version. In the following, I have two suggestions for the revised version.

The revision has added Sec. 3.6 to explain additional motivation. However, I wonder if the contents should be presented before explaining the proposal method, such as Sec. 1 or the first half of Sec. 3. Discussing motivation at the end of Sec. 3 makes the presentation feel like it is not progressing seamlessly. Also, it seems the response claims that the motivation is to save computational resources. Since it becomes a key claim of this paper, it will be better to show the results of computational efficiency in the experiments compared to the approach with separate models. Please note that the efficiency is not trivial: the resources for separate models could be smaller than the proposed joint approach for achieving enough performance in practice. Also, if the proposed method is running on an actual platform, I think the paper would be more convincing if this paper show the results of deploying it. If a method makes profits, showing those results is often seen as an influential contribution to the studies about online platforms (e.g., [4]).

The response often mentioned that the reason why some of the experimental results are missing is that those results were not so good and readers may be overwhelmed. I don't believe that not showing the results is an appropriate manner to report experiments. If this paper does not show the results, readers cannot know how good or bad the results are. Even if the results are not so good, reporting all the results will provide valuable information for subsequent studies. In the first place, readers will not be overwhelmed even if this paper shows the missing results. Bolding and underlining should be used for the readers who only want to know a summary of the results.

[4] McMahan et al., Ad Click Prediction: a View from the Trenches, KDD 2013.


**Summary Of The Paper:**

This paper proposes to address candidate generations of users and items for recommender and advertising systems simultaneously by a single proposed model. The proposed model provides two ideas for existing two-tower recommendation models: it introduces a normalization of the score function and a bidirectional version of the InfoNCE loss. This paper evaluated the proposed method through several experiments.


**Summary Of The Review:**

While this paper addresses an interesting problem, the significance is not sufficiently discussed in the paper. The novelty and effectiveness of the proposed method will be quite weak. The clarity needs to be improved.

---

> ### Author Response · Authors · 2021-11-22
> **Reply to Reviewer knEH**
>
> Thank you very much for your comments and suggestions. Our reply is as follows:
>
> 1.	Paragraph 1 of cons
>
> A: We apologize for that we might not have stated our motivations in depth in the previous draft. Please refer to "Reply to two common concerns: Motivation and Technical Contribution" for more details about the motivations.
>
> Theoretically, for a pure recommender system, modeling p(u,i) will not bring much gain of accuracy compared to modeling p(i|u). The increase of the diversity is the main benefits, but may not be the key metric considered by practitioners of the recommender systems.
>
> To the best of our knowledge, we didn't find any previous studies performing both recommendation and advertising simultaneously. Please kindly let us know if you happen to know.
>
> 2.	Paragraph 2
>
> A: About the technical contribution, please refer to our replies "A new theoretic discovery" and "Reply to two common concerns: Motivation and Technical Contribution".
>
> We are aware that "l2 normalization and temperature parameter" has been proposed in SimCLR. Our paper emphasizes mainly on their usefulness as a kind of regularization trick. One may not consider this as our main contributions.
>
> 3.	Paragraph 3
>
> A: We name the method "TotalRecall" because it can generate candidates from both directions. In theory, modeling p(u,i) [TR] will not increase the recall rate of u2i candidates generation compared to modeling p(i|u). The main reason for modeling p(u,i) can be found in our reply "Reply to two common concerns: Motivation and Technical Contribution", and we will update our draft accordingly.
>
> About the convergence speed, our explanation consists of three points:
>
> i). For MF_b[0:8], 9 processed samples consist of only 1 positive samples on average. While in Bi-InfoNCE or Uni-InfoNCE, 9 processed samples are all positive samples.
>
> ii). For MF_b, each sample is either positive or negative, and the binary-cross-entropy loss only allows it to compare with itself. For Bi-InfoNCE or Uni-InfoNCE, softmax-cross-entropy loss allows each sample can compare with all the other samples in the batch.
>
> iii). Compared Bi-InfoNCE to Uni-InfoNCE, Bi doubles the comparisons, and thus converges a bit faster than the Uni one.
>
> Bearing in mind that MF_b and Bi/Uni-InfoNCE can theoretically converge to the same optimum, we hope that our qualitative explanation can provide some intuitive understandings of the effectiveness of fast convergence.
>
> About the diversity of the Bi-InfoNCE compared to the Uni-InfoNCE, we draw this conclusion from the consistent experimental observations across different datasets. The only difference between Bi and Uni-InfoNCE is the modeling objective: p(u,i) versus p(i|u). We hypothesize that modeling p(u,i) will force the embedding space of u & i to be more coherent. But we don't have rigorous proof for this hypothesis.
>
> 4.	Paragraph 4
>
> A: About the HR and recall of the Uni-/Bi-InfoNCE and SSM, if evaluated with the metrics for u2i candidate generation, their difference is subtle in theory. Because if Uni-InfoNCE and SSM both model the p(i|u), the results shall be the same given that the model architectures are the same and both are well trained. Also, modeling p(u,i) will not gain much advantage if evaluated with u2i metrics HR and recall.
>
>
> We will address the remaining list of questions in a later reply. And we will update our draft accordingly.

---

> ### Author Response · Authors · 2021-11-23
> **Reply to some details 1**
>
> •	The notation 'N' is used many times with different meanings in the whole paper. I recommend consistently using it throughout the manuscript.
>
> Thank you for the suggestion. We will change our script accordingly.
>
> •	In the experiments, how many negative samples were used for all the methods in each experiment? I interpreted this paper used 100 negative samples in Sec. 4 according to the description of A.4.1. Did the TR methods also use 100 samples in Sec. 5? If so, it means that the TR methods used much larger samples than MF_b methods. Why?
>
> Thanks for the questions. I think you might mis-understand this part. The 99 negative samples are related to evaluation only.
>
> During training, MF_b has the same number of positive samples as TR (Bi-InfoNCE & Uni-InfoNCE), but MF_b samples negative samples explicitly (In Sec. 4, we sample 8 negative pairs for one positive pair), while TR use only positive samples. So in one epoch, the processed records of MF_b is 9 times of TR.
>
> TR uses other positive samples inside the batch as the “implicit” negatives to put into the softmax-cross-entropy loss.
>
> You may read more in our new draft in Sec. 4.1 on training efficiency.
>
>
> •	In the experiments, this paper refers too much to the figures in the appendix to explain the main experimental results. The appendix would be a material to support the main part, and I think it would be better to put the main results in the appendix much more sparingly.
>
> Due to the 9-page limit for the main body, we cannot put most of the content in the main body. We have tried our best to maintain the coherence of the content in the main body, while also put many results in the Appendix.
>
> •	In the experiments, why was the effect of different sizes of negative samples only evaluated in Sec. 5? It would be more consistent to do the same assessment in Sec. 4.
>
> In Sec.4, the baselines we compared to don’t consider the different sizes of negative samples. Actually, the size of negative samples is usually not of interest. The sampling methods are the key that determine the performance. This is shown in both the experiment results and theoretical derivation.
>
> •	In the experiments, please ensure what the bold and underline highlighting means for all tables. Table 4 appears to have duplicate underlines.
>
> Thank you. We will change our script accordingly.
>
> •	Sec. 1: this paper mentions 'Theoretically we show that the superiority is due to the modeling of joint probability of u and i.' However, I did not find any theoretical claims to show the superiority of the joint distribution.
>
> Here we mean that the better performance of our method originates from the modeling objective, i.e., p(u,i). Because we theoretically show that our \phi(u,i) converges to log{p(u,i)}, while \phi(u,i) of other methods converge either to log{p(i|u)} or log{p(u|i)} or log{p(i|u)/p(i)}.
>
> •	Sec. 2.3: I recommend specifying x and y of Eq. (1).
>
> Thank you. We will change our script accordingly.
>
> •	Sec. 3.1: what exactly is y_{u,t}? Is this an indicator variable of a specific item that was clicked on by u at time t?
>
> Yes.
>
> •	Sec. 3.2.2 and A.1: I did not know what the variable s_u indicates. According to the text, s_u is drawn from the multinomial distribution Mult(N_u, p_u), which means that s_u is a K-dimensional count vector where s_{u, i} (i=1, ..., K) denotes the i-th element of s_u and corresponds to the number of interactions between u and i. However, Eq. (5) refers to s_u as an indicator of item i. Also, if S_{u, i} is a binary matrix, s_u will result in a binary vector. As in (Liang et al., 2018), the observation of s_u may happen to be binary, but this paper fixes the variable to be binary in the problem definition of Sec. 3.1.
>
> Thank you very much for pointing out! We will change our script accordingly.
>
> In the dataset used by (Liang et al., 2018), e.g., movie-lens dataset, the observation of s_u is also fixed to be binary.
>
> Actually, we can also extend to non-binary scenario without additional modifications.

---

> ### Author Response · Authors · 2021-11-23
> **Reply to some details 2**
>
> •	Sec. 3.4.1: I did not understand why this paper chose a classical two-tower architecture. It seems that the presented two reasons do not exclude other possible architectures.
>
> Because in large-scale RS or PUMS, the whole process is divided into two stages roughly: candidate generation (CG) stage and Ranking stage. For example, in CG of RS, we need to retrieve thousands of items from the billion-scale item-pool for each user.
>
> So, we need to disentangle users’ and items’ embeddings, and predict them separately. During the online serving, we can use ANN search (such as Faiss) to retrieve the top items for each user. ANN search can calculate the dot product of users’ and items’ embeddings, and find the approximate nearest items for each user very fast. [Refer to Youtube DNN paper]
>
> •	Sec. 3.4.1: 'in online serving .' -> 'in online serving.'
>
> Thanks. We will change our script accordingly.
>
> •	Sec. 3.4.1: the notation of f_{\theta}(u) is confusing since it quite overlaps the notation of f_{\theta}(u, i) in Eq. (1). Also, f_{\theta}, g_{\theta}, and \phi_{\theta} are also confusing since they share the same parameter \theta. I recommend using different variables or specifying the detail of \theta.
>
> Thanks for the suggestion. We will change our script accordingly. However, for f_{\theta}, g_{\theta}, and \phi_{\theta} in Sec. 3.4.1, they refer to the different parts of the same model, so we will keep it.
>
> •	Sec. 3.4.1: \phi_{\theta} is used for both unnormalized and normalized score functions, which are confusing a little. For both functions, it is not clear what \theta is.
>
> We will change the notations for unnormalized and normalized score functions.
>
> \theta is the parameters of the two towers.
>
> •	Sec. 3.4.2: the definition of |S_{u, i}| seems unclear in Eq. (3). S_{u, i} is defined as a matrix, and the notation of |S_{u, i}| is not specified.
>
> Thanks for pointing out. |S_{u, i}| means the sum of the entries of matrix S.
>
> •	Sec. 4.1: 'NCF' of the caption of Figure 2 is not defined in this paper.
> Thanks for pointing out. We will change our script accordingly.
>
> •	Sec. 4.1 and Sec. 5.2: this paper claims the TR methods converge faster than compared methods from the results of Fig. 3 and Fig. 7, but their x-axis denotes the number of processed data samples. I recommend showing the results with actual runtime on the x-axis.
>
> The model architectures are the same for TR and other baseline models, and the loss calculations of InfoNCE (TR) and binary-cross-entropy (MF_b) are not the bottleneck, so we choose the number of processed data samples as x-axis. The actual runtime is often not stable and can fluctuate.
>
> Other papers like CLIP by OpenAI also use the number of processed data samples as x-axis for comparing convergence speed.
>
> •	Sec. 4.1: it would be better to explain what distinct items are before introducing Fig. 3. The term 'distinct items' is first explained in A.4.2.
>
> Thanks for pointing out. We will change our script accordingly.

---

> ### Author Response · Authors · 2021-11-23
> **Reply to some details 3**
>
> •	Sec. 4.2: I didn't understand how to make the data for next-n-item prediction, which should be explained in Sec. 3.1. Also, this paper should ensure that using such data leads to fair comparison for all methods.
>
> We only compare next-n with next-1 for the same method to show the usefulness of next-n. So we didn't apply for all methods.
>
> •	Sec. 4.2: why did not the experiment report the NDCG as in (Cen et al., 2020)?
>
> Because their NDCG calculate has defect, and the author advise to use HR and recall in his git.
>
> •	Sec. 4.2: why did not Table 2 report the result of 'TR-Youtube [next-7-prediction]' for the Taobao dataset?
>
> Because the experiment and analysis of distribution p(y) show that in this dataset, next-n does not bring gains.
>
> •	Sec. 4.2: how does each method in Fig. 4 correspond to the method in Table 2? Also, I did not fully understand the meaning of '_temp' in the caption of Fig. 4.
>
> '_temp' means the l2-normalizing the embeddings and rescaling with the \tau. And we already explain it in the figure capation.
>
> •	Sec. 4.2: what is the coherent vector? How is it induced by Eq. (3) and how does it improve diversity?
>
> Thanks for the question. We will change our script accordingly.
>
> •	Sec. 5: in the first paragraph, please provide more specific examples that motivate mining for a group of similar items in real E-commerce services.
>
> Thanks for the suggestion, we will change our script accordingly.
>
> •	Sec. 5: in the first paragraph, this paper mentions 'Previously, ... so many models.' However, in the experiment of Sec. 5, only 54 item groups were used. I think binary classification approaches are also applicable for item groups.
>
> Yes. But this will require us to prepare 54 training data, train 54 models, and make prediction 54 times. This method is not scalable in practice, although it is widely used when we mine potential users for only several groups.
>
> •	Sec. 5.1: this paper mentions that 'We use Movielens-1m because it is also used in MFb of RS.' However, the Pinterest dataset was also used in the experiment of Sec. 4. Why was only the Movielens-1m dataset used?
>
> Because in the Pinterest dataset, the categories of items are not provided. We trace back to the raw data of Pinterest, and didn’t find the categories so that we can do candidate generation for a group of items.
>
> •	Sec. 5.2: this paper mentions 'TR uses the same training data and model configurations for both RS and PUMS.' However, the procedure for creating the datasets appears to be different in Sec. 4 and Sec. 5. I didn't know if this paper used the same TR models for both experiments.
>
> Yes, the procedure for creating the datasets is different. This is because in Sec.4, we have to follow the procedure of previous papers to do a fair comparison. But the data splitting is specific for RS, but not suitable for PUMS, so we have to process the data differently.
> After the data splitting, the remaining data processing procedures are the same, for example, the calculation of empirical marginal distribution p_data(u) and p_data(i).
>
> The hyperparameters like \tau and batch-size are slightly different because the training data is not exactly the same.
> In practical application (See our reply about the “Motivations”), u2i and i2u are applied together with the u/i embeddings from the one trained TR model.

---

> ### Author Response · Authors · 2021-11-23
> **Reply to some details 4**
>
> •	Sec. 5.2 and Sec. 5.3: I recommend showing all the scores of the distinct users even if the accuracies are very low.
>
> Thanks for pointing out. We think that readers might be overwhelmed if too many numbers are presented and not significant for our purpose.
>
> •	Sec. 5.3: 'For small dataset' -> 'For small datasets'?
>
> Thanks for pointing out. We will change our script accordingly.
>
> •	Sec. 5.3: I recommend showing all the results of aggregation methods: several results with max and sum are missing in Table 4.
>
> Thanks for pointing out. Same as our previous concern, readers might be overwhelmed if too many numbers are presented and not significant for our purpose and results.
>
> •	A.1: did s_{u, i} follow the Gauss distribution even if it is a binary random variable?
>
> Yes. One can assume that s_{u, i} follows the Gauss distribution, even if it might lead to bad results given that it is actually binary.
>
> •	A.2: S -> S_{u,i}?
>
> Thanks for pointing out. We will change our script accordingly.
>
> •	A.2: what is exactly N in Eq. (8)? Intuitively, I think Eq. (8) should be divided by M * K.
>
> Thank you very much for pointing out. We will change our script accordingly.
>
> Actually, N is the total number of samples in the training data, including both positive samples and negative samples.
>
> We have changed the notion in Eq. (8) to avoid the confusion.
>
> •	A.3: I did not understand the claim of this section. I recommend providing more formal proofs to show the convergence of the score function.
>
> Thanks for the suggestion. We will change our script accordingly.
>
> •	A.4: I think this section is not well-structured. Although the titles of subsections indicate the considered methods, they explain the evaluation metrics.
>
> Thank for the suggestion. We will change our script accordingly.

---

### Official Review · Reviewer_eQ3i · 2021-11-02

**Correctness:** 3
**Technical Novelty And Significance:** 2
**Empirical Novelty And Significance:** 2
**Recommendation:** 3
**Confidence:** 4

**Main Review:**

Strengths:
1. The paper is well-organized and clearly-written. The main idea is clear and easy to understand.
2. The proposed Bi-InfoNCE loss helps to model the joint distribution for bidirectional candidates generation.
3. The authors conduct experiments on several public datasets.

Weakness:
1. The idea of introducing InfoNCE loss to recommender system is not novel, which has been used in existing works. The main contribution of this paper is the design of Bi-InfoNCE loss to optimize the joint distribution $p_{data}(u,i)$. However, it is not demonstrated why modeling $p_{data}(u,i)$ would be better than separately modeling $p_{data}(u|i)$ and $p_{data}(i|u)$ for u2i or i2u candidate generation. From the results in Table 2, Uni-InfoNCE u2i performs even better than Bi-InfoNCE, which further weakens the motivation of the Bi-InfoNCE loss.
2. Besides, in Table 2, ComiRec methods perform significantly better than the proposed methods for Taobao dataset, but no discussion is provided. In this way, it is not approporiate to say ''TR achieves much better results by a large margin''.
3. The comparison results to MF in Figure 2 and Table 1 are not promising. TR performs better for Pinterest but MF achieves better HR for Movielens. Results at other truncated top values are not provided. The overall improvement over MF with point-wise loss is not significant.

**Summary Of The Paper:**

This paper proposes a method for both user2item and item2user candidates generation in recommender systems. The main idea is to employ infoNCE loss for optimizing the two tower model. To achieve bidirectional candidates generation, the authors propose to do negative sampling in batch from both user side and item side, and use the Bi-InfoNCE loss for optimization.

**Summary Of The Review:**

The advantage of the proposed Bi-InfoNCE loss is neither clear demonstrated nor well supported by the experimental results. Thus I tend to reject this paper.

---

> ### Author Response · Authors · 2021-11-22
> **Reply to Reviewer eQ3i**
>
> Thank you very much for your careful reading and comments. Our reply is as follows:
>
> 1. InfoNCE loss in Recommender System.
>
> A: Please refer to our reply "Reply to two common concerns: Motivation and Technical Contribution".
>
> 2. Table 2, ComicRec's good performance.
>
> A: We are sorry that we didn't explain it clearly. For a fair comparison, we shall only compare to the baseline "Youtube DNN", because their model architectures and capacities are the same. For ComiRec models, the architectures are much more complicated because they use capsule networks or attention mechanisms to obtain multiple user vectors (multi-interests).
>
> We keep the ComiRec's results because of the Amazon Books data. We want to show that, even with more interest vectors, if we don't regularize the model correctly (e.g., l2-normaling and re-scaling), the results can be worse than a base model architecture.
>
> 3.	comparison results to MF in Figure 2 and Table 1.
>
> A: Thank you for pointing out. We might emphasize wrongly about the performance gains from TR. Actually, the performance is very close, and our new theoretic discovery (refer to our reply "A new theoretic discovery") shows that these modeling objectives are equivalent.
>
> Because we use the same model architecture (i.e., the parameters in two towers are the same), so their performance shall be close. The minor difference could be due to the training configurations, for example, regularization strength, optimization algorithms and etc.
>
> The largest gain of TR is closely related to our motivations and applications (refer to "Reply to two common concerns: Motivation and Technical Contribution"). Besides, TR can ensure fast convergence and diversified results while keeping the performance compared to Uni-InfoNCE and MF_b.
>
> We will modify our draft to address these concerns.

---

### Official Review · Reviewer_vmLg · 2021-11-06

**Correctness:** 2
**Technical Novelty And Significance:** 2
**Empirical Novelty And Significance:** 2
**Recommendation:** 3
**Confidence:** 4

**Main Review:**

++ It is a good try to consider RS and AS together.

However, there are major flaws in the current forms of the paper.

-- The motivation is not strong. Why we should consider RS and AS jointly; what are the benefits and what are the challenges. Motivation should be stated more clearly.

-- The technical contribution is not enough. Comparing the objective of the proposed model (see Eq. (3)), it is a direct expansion of the existing loss functions (see Eq. (1)) from uni-loss to bi-loss (user & item).

Smaller ones:

-- The organization and writing can be improved. The mathematical notations are not correct in some places (both in the body text and in the Appendix).

-- For evaluating advertising systems, the MovieLens dataset is not a suitable one and the CTR (click-through rate) metric should be reported.

-- many typos:
In Abstract: Amazaon -> Amazon, generation compare to strong baselines -> generation compared to strong baselines.
In Contributions: with higher accuracy compare to other methods -> with higher accuracy compared to other methods.
Lines before Eq. (2): find that l2-normalize u and i and then rescale the dot product -> find that l2-normalizing u and i and then rescaling the dot product.


**Summary Of The Paper:**

This paper jointly considers the problems of recommender systems (RS) and advertising systems (AS) as it is a coin’s two sides. The key is to model the joint probability of (user, item) instead of conditional probabilities (user | item) and (item | user) independently. The proposed model adapts an existing large-vocabulary loss function to the optimization objective. Experiments on real-world datasets are conducted to compare with both Matrix Factorization and Sequential Modeling methods.

**Summary Of The Review:**

Lacking motivation, contribution weak, evaluation not solid.

---

> ### Author Response · Authors · 2021-11-22
> **Reply to Reviewer vmLg**
>
> Thank you for your comments and suggestions of our work. Our reply is as follows:
>
> Major
>
> 1.	Motivation
>
> A: Please refer to our reply "Reply to two common concerns: Motivation and Technical Contribution".
>
> 2.	Technical contribution
>
> A: Please refer to our reply "A new theoretic discovery".
>
> Minor
>
> A: Thank you for pointing out these concerns.
>
> For evaluating PUMS (advertising systems), CTR is usually not the metric to be used. CTR is usually adopted in the ranking stage of the Recommender System or Advertising System. In candidate generation of PUMS, we usually care about the precision and recall.
>
>
> Thank you again for your comments. We hope that our reply addresses your concerns. We will update our draft accordingly.

---

### Author Response · Authors · 2021-11-21
**A new theoretic discovery**

Thank all the reviewers for their comments and questions.

Before replying to the questions, we want to present a new theoretic discovery: we find that the two modeling strategies (MF with Bernoulli distribution and Multinomial distribution) are equivalent. Specifically, depending on the different negative sampling methods of MF_b, the dot product \phi(u,i) of MF_b will approximate one of the four: log{p(i|u)}, log{p(i|u)}, log{p(u,i)/[p(u)p(i)]}, and log{p(u,i)}.

MF_b[0:8] -> p(i|u), MF_b[8:0] -> p(u|i), MF_b[4:4]/MF_b[8:8] -> p(u,i)/p(u)p(i) == p(u|i)/p(u) == p(i|u)/p(i) [CLRec's modeling objective], MF_b[uniformly sampling all u-i pairs as negatives] -> p(u,i) [our TR's modeling objective].

The theoretic proof is inspired by Gutmann's NCE paper, and can be found in Sec. 3.5 and App. A.4 of the newly uploaded draft.

In the ranking stage of industrial recommender systems, people usually model the single u-i pair and assume that it follows Bernoulli distribution, and in the candidate generating stage, people assume that user-clicked items follow Multinomial distribution. To the best of our knowledge, we are the first to show that they are theoretically equivalent.

---

### Author Response · Authors · 2021-11-22
**Reply to two common concerns: Motivation and Technical Contribution**

Now we will address two common concerns of reviewers: motivation and technical contribution.

1. Motivation.

A: TR is developed to solve two practical problems: one is the large-scale potential users mining, and the other is the marketing in small and medium enterprises (SMEs).

    - For example, we have 1 billion users and 10,000 items, and we need to mine the top 1 million users for each of the items. Here we list all the possible methods we can think of:
        a). For each item, we create the training sample, and then train a binary classification model, and finally predict for the 1 billion users and pick the top 1 million. We need to repeat 10,000 times.
        b). By emulating the recommender system, we model p(u|i) and obtain the embeddings of users and items, and then apply ANN search (e.g., Faiss) to find the top 1 million users for each item. This method has a serious defect: ANN has high accuracy if we retrieve the top hundreds or thousands of candidates, but its accuracy and efficiency drops dramatically if we extend the range to millions of candidates. So, this method is not practical.
        c). Also, we model p(u|i) and obtain the embeddings of users and items, and then do a brute force search: we calculate the similarity score of all the users and items, which is the Cartesian product of 1 billion users and 10,000 items, i.e., 10 trillion, and then we pick the top 1 million users for each item. It's very inefficient and consume huge number of resources.
        d). We build two models for p(i|u) and p(u|i) respectively. Then we use the embeddings of p(i|u) and ANN search to find top 10 items for each user. Now we have 10 billion u-i pairs, and we use the embeddings of p(u|i) to calculate the similarity score of the u-i pairs, and finally pick the top 1 million for each item.
        e). We model p(u,i) with TR. Then we use the embeddings and ANN search to find top 10 items for each user. Now we have 10 billion u-i pairs and their similarity score, and we just pick the top 1 million for each item based on the same score.

Method e) has just one less model than d) and one less step, but in practice, it is problematic to maintain two correlated models, especially when they are very large. Two models usually mean the calculation is doubled, the storage is doubled, and the risk and instability are doubled.

    - SMEs usually have several channels to sell their products, for example, e-commerce platforms like Amazon and Alibaba, their own websites, social medias like Facebook and Instagram, and offline shops.
    They usually need to market their products in two ways: a). SMEs pick some active users or in-active users from different channels, and then choose the topN products for each user and send the personalized campaign messages to them. This can be regarded as the recommender system. b). SMEs has some new or special products and want to find the target users. This is potential users mining.
    The two marketing requirements can be complete either with two separate models or with one unified model (TR). Due to the limited computation and storage budgets, and shortage of talents, TR is the best choice for them. Actually, TR is already implemented in one of our cloud-service products for SMEs.

To conclude, we may not state clearly about the motivation of TR, but it can really solve our practical problems and is already deployed in our production systems and cloud-service products for SMEs. The deployed model has much more complicated tower architectures, but we think it is not the key point and does not discuss it in the paper.

2. Technical Contribution

A: Although Bi-InfoNCE is a direct expansion of Uni-InfoNCE, we rigorously prove that the \phi(u,i) converges to the logarithm of the joint probability. Actually, SimCLR and CLIP also used the bi-directional InfoNCE loss, but they used it just for symmetry consideration and fast convergence, and didn't connect it with the joint probability [p(u,i)] or point-wise mutual information [p(u,i)/(p(u)p(i))].

Our practical applications (motivations) require that we should have a clear understanding of \phi(u,i). In contrast, SimCLR and CLIP only care about the learned embeddings. So, the Bi-InfoNCE might just be a minor contribution for SimCLR and CLIP, but its physical meaning is the foundation in our applications.

What's more, after we made the final submission, we re-think about the two modeling objectives: Bernoulli distribution and Multinomial distribution. Because their results are so close, we tried to connect them and finally prove that they are equivalent.

To conclude, we hope that our explanations and the new discovery can show that our technical contribution is novel and useful in our applications, and also benefit professionals in this and related research areas.

Later we will upload a new version of draft and reply to each reviewer separately.

---

### Decision · Program_Chairs · 2022-01-20

**Decision:**

Reject

**Comment:**

Reviewers are in agreement that the paper is below the acceptance threshold. Main concerns focus around novelty, experiments, and justification of the paper's main claims.